# Unlocking Efficient, Scalable, and Continual Knowledge Editing with Basis-Level Representation Fine-Tuning

**Tianci Liu**[1], **Ruirui Li**[2], **Yunzhe Qi**[3], **Hui Liu**[2], **Xianfeng Tang**[2], **Tianqi Zheng**[2], **Qingyu Yin**[2], **Monica Cheng**[2], **Jun Huan**[4], **Haoyu Wang**[5], **Jing Gao**[1]
[1]Purdue University   [2]Amazon   [3]UIUC   [4]AWS AI Lab   [5]SUNY Albany
[1]{liu3351,jinggao}@purdue.edu   [2]ruirul@amazon.com   [5]hwang28@albany.edu

## ABSTRACT

Large language models (LLMs) have achieved remarkable performance on various natural language tasks. However, they are trained on static corpora and their knowledge can become outdated quickly in the fast-changing world. This motivates the development of knowledge editing methods designed to update certain knowledge in LLMs without changing unrelated others. To make selective edits, previous efforts often sought to update a small amount of parameters in some specific layer(s) of a LLM. Nonetheless, in challenging scenarios, they still fall short in making successful edits while preserving knowledge irrelevant to the updates simultaneously, resulting in a notable *editing-locality* trade-off. In this work, we question if the trade-offs are caused by the fact that parameter-based updates have a global effect, i.e., edited parameters affect all inputs indiscriminately. In light of this, we explore the feasibility of representation fine-tuning, which applied some linear update to a few representations in a learned subspace, for knowledge editing. While being effective to enhance an LLM's general ability as demonstrated in the previous work, we theoretically show that this linear update imposes a tension in editing-locality trade-off. Subsequently, BaFT is proposed to break the linearity. BaFT computes a weight for each basis that spans a dimension of the subspace based on the input representation. This input-dependent weighting mechanism allows BaFT to manage different types of knowledge in an adaptive way, thereby achieving a better editing-locality trade-off. Experiments on three LLMs with five editing benchmarks in diverse scenarios show the superiority of our method.

## 1 INTRODUCTION

Language models (LMs) parameterized by deep neural networks (Vaswani et al., 2017; Lewis et al., 2019; Radford et al., 2019; Brown et al., 2020) have thrived in producing fluent and meaningful texts on diverse natural language generation and classification tasks (See et al., 2019; Raffel et al., 2020; Ji et al., 2023). These successes underscore the versatility of LMs, establishing them as the foundations for different natural language processing applications (Bommasani et al., 2021; Zhou et al., 2023). Additionally, with model sizes continually increasing, large language models (LLMs) have demonstrated unprecedented abilities to follow natural language instructions (Dong et al., 2022b; Ouyang et al., 2022), empowering zero-shot adaptations to unseen tasks (Kojima et al., 2022), and paving the way towards artificial general intelligence (Bubeck et al., 2023).

Despite their remarkable performance, the real-world deployment of LLMs remains largely unresolved. While LLMs can understand a wide range of contexts, they can only provide feedback based on the *static* knowledge from the data on which they were trained. In a fast-changing world, most knowledge quickly becomes outdated. This could amplify critical issues such as making factual fallacy (De Cao et al., 2021) or producing harmful generations (Hartvigsen et al., 2022).

As a remedy, *knowledge editing*, whose goal is to update an LLM with some specific new knowledge without hurting irrelevant knowledge, has been proposed (Wang et al., 2023; Zhang et al., 2024b). Early effort of full fine-tuning proved ineffective as it also disrupted irrelevant knowledge (Wang

et al., 2023), leading to an *editing-locality* trade-off. Here *locality* refers to the ability to maintain the knowledge that is irrelevant to the updates. To achieve a good locality, the model update needs to be *selective* and should rely on a small number of parameters (Wang et al., 2023), and thus parameter-efficient fine-tuning (PEFT) methods like AdaLoRA (Zhang et al., 2023) have shown good performance (Wu et al., 2023). On the other hand, Huang et al. (2023); Dong et al. (2022a) restricted updates to specific feed-forward network (FFN) layer that served for knowledge storing (Dai et al., 2021). Meng et al. (2022a;b) refined the process through a *locate-and-edit* paradigm which involves an additional *locating* stage to identify which layer the target knowledge is stored. Nonetheless, these methods still exhibit a certain editing-locality trade-off, regardless of whether locating is performed. We note that these methods are parameter-based and have a global effect, i.e., the edited parameters affect *all* inputs indiscriminately. This observation challenges to what extent an editing can truly benefit from the targeted effort to identify "better" parameters that "memorize" certain knowledge (Hase et al., 2024). In other words, it is an open question *if such trade-offs are due to the coarse control of global parameter-based updates.*

This paper, following Hernandez et al. (2023) that modifies LLM knowledge by updating representations, explores *selective* representation-based knowledge editing, and paves a way for an affirmative answer to the above question. Our work is built upon ReFT (Wu et al., 2024) that fine-tunes a few representations in a low-rank linear subspace, and performs on par with PEFT methods such as LoRA family (Hu et al., 2021; Zhang et al., 2023; Ding et al., 2023) and others (Houlsby et al., 2019; Chen et al., 2024). Unlike parameter-based updates that apply to all inputs, ReFT only alters representations at some locations. Consequently, ReFT can achieve a better editing-locality trade-off than parameter-based updates. Notwithstanding, in spite of this promising result, the *subspace-level linearity* still restricts ReFT from providing precise enough updates for knowledge editing.

Specifically, ReFT applies the linear update in the subspace for *all* selected representations. While being effective to enhance an LLM's general ability such as commonsense reasoning (Wu et al., 2024), this subspace-level control can be too coarse for knowledge editing. As a consequence, when ReFT achieves high editing performance, certain unrelated knowledge may be modified incorrectly, provably jeopardizing its locality. This insight is formalized in Sec 2, where a theoretical analysis on this inherent tension is derived, based on two reasonable assumptions on how representations convey different knowledge. Notably, *our analysis reveals an intrinsic limitation of linear representation fine-tuning. It not only holds for knowledge editing, but also applies to other tasks that require selective updates such as continual learning and machine unlearning, and can be of independent interest to these communities.* This theoretical result is one of the main contributions of this paper.

In light of this insight, we derive BaFT, a more precise representation fine-tuning method for knowledge editing. Noting that the subspace is spanned by a group of bases vectors, BaFT instead learns a *basis-level* update. This involves computing a weight for each basis for a given representation, then learning a linear update along this basis. Since each basis spans a rank-1 subspace, BaFT is a generalization of ReFT, in the sense that if all bases use the same constant weight 1, BaFT reduces to ReFT. By using different weights combinations on distinct types of knowledge, BaFT can manage them in a more adaptive way. When auxiliary locality information (e.g., what knowledge should *not* be updated) is available, BaFT can freely restrict the impact of unimportant bases only, while ReFT needs to regulate the whole subspace rigidly. This flexibility makes BaFT highly suitable for knowledge editing and performs on par with the strongest baseline that relies on external memories to memorize new knowledge and requires 10-20 times more parameters. In conclusion, *BaFT, as a new representation fine-tuning method, successfully reaches a better editing-locality trade-off while maintaining the parameter efficiency of ReFT.* This is another main contribution of this work.

Our paper is organized as follows. Sec 2 details the proposed BaFT. Extensive experimental results in Sec 3 demonstrate the superiority of our method for conducting knowledge editing at much better parameter efficiency than existing methods. In the remaining part of this paper, we review related works in Sec 4, and conclude the paper in Sec 5.

## 2 PROPOSED METHOD

Grounded in a theoretical analysis, we show that the linearity nature of existing representation fine-tuning method induces an inherent limitation on its editing-locality trade-off. We then propose BaFT towards a fine-grained controlled representation fine-tuning in accordance with knowledge editing.

## 2.1 PRELIMINARIES

Given input $\boldsymbol{x} = (x_1, \ldots, x_n)$, where each $x_i \in \mathcal{V}$ is a token from vocabulary $\mathcal{V}$, a language model (LM) parameterized by $\theta$ assigns probability $p_\theta(\boldsymbol{x})$ using the chain rule (Bengio et al., 2000):

$$p_\theta(\boldsymbol{x}) = \prod_{i=1}^{n} p_\theta(x_i \mid x_1, \ldots, x_{i-1}) \triangleq \prod_{i=1}^{n} p_\theta(x_i \mid \boldsymbol{x}_{<i}),$$

where $p_\theta(x_i \mid \boldsymbol{x}_{<i})$ is the predicted distribution of the next token $x_i$ over $\mathcal{V}$ given previous $\boldsymbol{x}_{<i}$. In specific, for an $L$-layer LM, let $\boldsymbol{h}_i^{(l)}$ denote the intermediate *representation* of the $i$-th token at the $l$-th layer. The predicted distribution is given by softmax regression parameterized by $\mathbf{W}$ at layer $L$:

$$p_\theta(x_i \mid \boldsymbol{x}_{<i}) = \text{softmax}(\mathbf{W}\boldsymbol{h}_i^{(L)}).$$

To generate a sentence $\boldsymbol{x}$, the LM repeatedly computes $p_\theta(x_i \mid \boldsymbol{x}_{<i})$ and draws $x_i$ from it; then $x_i$ is fed back into the LM as part of the inputs for future steps. The generation process completes if a special token that marks the end of the sentence is returned, or the maximum length is reached.

**Knowledge Editing** aims to incorporate new provided knowledge into a pre-trained LM while preserving other existing knowledge that shouldn't be modified. Formally speaking, any knowledge can be represented in natural language with a textual pair $(\boldsymbol{x}, \boldsymbol{y})$, where $\boldsymbol{x}$ entails some *subject* and *relation*, and $\boldsymbol{y}$ refers to the corresponding *object*. For instance, given $\boldsymbol{x}$ being *The current president of United States is*, $\boldsymbol{y}$ can be *Joe Biden*. Knowledge editing seeks to maximize the chance of an LLM responding with $\boldsymbol{y}$ given $\boldsymbol{x}$, while satisfying the following additional criteria at the same time (Zhang et al., 2024b; Liu et al., 2025a): (1) **Generality:** there are different ways to express *US president*, wherefore the edited model should generalize. (2) **Portability:** relevant knowledge such as *the first lady of United States* should be updated as well. (3) **Locality:** irrelevant knowledge such as *the prime minister of United Kingdom* should not be affected. Notably, such requirements of modifying only specific internal knowledge in a LM has been proved challenging. As revealed in previous works (Zhang et al., 2024b), this process should update only a minimal amount of parameters.

**Representation Fine-tuning (ReFT)**, proposed by Wu et al. (2024), is a recent parameter-efficient fine-tuning (PEFT) method that outperformed other approaches such as LoRA in updating pre-trained LM on several tasks with much less parameters. Building upon the so-called *linear representation hypothesis* (Park et al., 2023) which presumes that *concepts are encoded in linear subspace of representations*, ReFT learns *low-rank linear updates* on representations. In particular, to update the $d$-dimensional representation $\boldsymbol{h}_i^{(l)}$ at layer $l$ for the $i$-th token, ReFT learns

$$\Phi_l(\boldsymbol{h}_i^{(l)}; \phi_l) = \boldsymbol{h}_i^{(l)} + \mathbf{R}_l^\top(\mathbf{A}_l \boldsymbol{h}_i^{(l)} + \boldsymbol{b}_l - \mathbf{R}_l \boldsymbol{h}_i^{(l)}), \tag{1}$$

where $\phi_l = (\mathbf{R}_l, \mathbf{A}_l, \boldsymbol{b}_l)$ are learnable parameters added to layer $l$. Here $\mathbf{R}_l \in \mathbb{R}^{r \times d}$ is a low-rank matrix (i.e., $r \ll d$) containing mutually orthogonal rows that specifies a subspace to make the update, and $(\mathbf{A}_l, \boldsymbol{b}_l)$ predicts the *updated* representation in this subspace. Finally, ReFT requires hyper-parameter $I \subset [n]$ to specify which locations need updates. Put together, ReFT intervenes the layer $l$'s output by

$$\boldsymbol{h}_i^{(l)} \leftarrow \left( \Phi_l(\boldsymbol{h}_i^{(l)}) \text{ if } i \in I \text{ else } \boldsymbol{h}_i^{(l)} \right)_{i \in 1, \ldots, n}.$$

From now on, we omit indices $i, l$ when discussing how a representation $\boldsymbol{h}$ is intervened for brevity.

## 2.2 EDITING KNOWLEDGE BY FINE-TUNING REPRESENTATIONS

ReFT has demonstrated impressive performance on tasks such as commonsense reasoning that largely rely on an LLM's ability to understand and generate text by updating *just a few* (i.e., those in $I$) representations. However, it is unknown whether this lightweight approach can benefit knowledge editing, which requires modifying some selective internal knowledge. Here, we show that the linearity nature of ReFT limits its editing and locality performance. In specific, for all inputs, ReFT applies the *same* linear update without distinction:

$$\Phi(\boldsymbol{h}) = \boldsymbol{h} + \mathbf{R}^\top(\mathbf{A}\boldsymbol{h} + \boldsymbol{b} - \mathbf{R}\boldsymbol{h}) = \underbrace{(\mathbf{I} + \mathbf{R}^\top(\mathbf{A} - \mathbf{R}))}_{\text{weight}} \boldsymbol{h} + \underbrace{\mathbf{R}^\top \boldsymbol{b}}_{\text{bias}}.$$

The coarse control from the linear ReFT makes it less suitable for knowledge editing for two reasons.

First, ReFT uses its learned subspace for editing in a prede-termined manner, regardless of varying levels of learning difficulty for different types of knowledge. This can lead to sub-optimal performance. As an evidence, we fit a rank-12 sub-space for ReFT and checked how many dimensions (bases) contribute negligible updates, as a measure of *dimension redundancy*. To this end, we count for each dimension, if its update magnitude is less than $M$ times of the maximal dimension. Fig 1 shows these results. We noted that the dimension redundancy indeed varies on different types of knowledge.

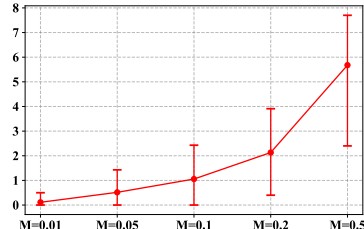

Figure 1: Averaged (w/ max-min range) number of *redundant* dimensions (which have update $M$ times smaller than maximal values), in a rank-12 ReFT update.

Second, the linearity of ReFT leads to an inherent editing-locality trade-off: it is challenging to maintain good *generality* and *locality* at the same time. Formally, given some knowledge involves subject $s$, relation $r$, and object $o$ *that can be updated by ReFT*, we make the following assumptions.

**Assumption 2.1.** Let text $\boldsymbol{x}$ encodes $s, r$. Since the knowledge can be edited by ReFT, text $\boldsymbol{y}$ generated by the LM will convey $o$ if its intermediate representation takes some targeted value $\boldsymbol{t}$.

**Assumption 2.2.** (Hartvigsen et al., 2024) For any $\boldsymbol{h}$ carrying some knowledge, there exists a positive $\varepsilon(\boldsymbol{h})$-radius $\ell_2$ ball $B(\boldsymbol{h}, \varepsilon(\boldsymbol{h}))$ around $\boldsymbol{h}$ such that any $\boldsymbol{h}' \in B(\boldsymbol{h}, \varepsilon(\boldsymbol{h}))$ conveys the same knowledge, we refer to $B(\boldsymbol{h}, \varepsilon(\boldsymbol{h}))$ as a *stable-ball* of $\boldsymbol{h}$.

We provide a few clarifications on the two assumptions. The first assumes that a piece of knowledge can be generated (retrieved) from some associated representation. The second, as in Hartvigsen et al. (2024), assumes that the knowledge is locally stable around its representation, so that a small perturbation won't change the carried knowledge. Under these two assumptions, The following Thm 2.3 reveals a tension between maintaining good *generality* and *locality* simultaneously, with its proof deferred to App B.1.

**Theorem 2.3.** *When fine-tuning an LM, ReFT learns to update the old representation $\boldsymbol{h}_0$ to targeted $\boldsymbol{t} = \Phi(\boldsymbol{h}_0)$. If ReFT maintains good generality such that $\forall\, \boldsymbol{h} \in B(\boldsymbol{h}_0, \varepsilon(\boldsymbol{h}_0))$,*

$$\|\Phi(\boldsymbol{h}) - \Phi(\boldsymbol{h}_0)\| = \|\Phi(\boldsymbol{h}) - \boldsymbol{t}\| < \varepsilon(\boldsymbol{t}),$$

*where $\|\cdot\|$ denote the $\ell_2$ norm. Then for any irrelevant input $\boldsymbol{h}_{ir}$ with a small stable-ball radius*

$$\varepsilon(\boldsymbol{h}_{ir}) < \frac{\|\boldsymbol{t} - \boldsymbol{h}_0\| - (\varepsilon(\boldsymbol{t}) + \varepsilon(\boldsymbol{h}_0))}{\varepsilon(\boldsymbol{t}) + 2\varepsilon(\boldsymbol{h}_0)} \varepsilon(\boldsymbol{h}_0),$$

*and is not too far from $\boldsymbol{h}_0$ such that*

$$\|\boldsymbol{h}_{ir} - \boldsymbol{h}_0\| = \varepsilon(\boldsymbol{h}_{ir}) + \varepsilon(\boldsymbol{h}_0),$$

*ReFT will output $\Phi(\boldsymbol{h}_{ir}) \notin B(\boldsymbol{h}_{ir}, \varepsilon(\boldsymbol{h}_{ir}))$ and break its locality guarantee.*

Intuitively speaking, Thm 2.3 formalizes that ReFT update has to be *large* enough to make successful edit; and *smooth* enough to achieve good generality. Then, due to its linearity, it will inevitably hurt the locality of some irrelevant knowledge. This limitation does not rely on the specific $r$ (i.e., subspace rank) being used. In summary, ReFT is less suitable for knowledge editing because of the two limitations, which motivates BaFT as presented in the next section.

## 2.3 BaFT: Basis-Level Representation Fine-Tuning

Given the two limitations from *linearity*, i.e. using the whole linear *subspace* to update all representation without distinction, and the finding of dimension (basis) redundancy, we propose to take the importance of each dimension into account. Since in ReFT, the subspace is parameterized by a set of orthogonal bases vectors, we assign each *basis* a *learnable weight* to determine how much it contributes to the current editing. This input-dependent weighting mechanism makes our method applies a non-linear update. We dub our method *ba*sis-level representation *f*ine-*t*uning (BaFT).

To be more specific, at a layer where ReFT takes place, we learn an $r$-dimensional update by

$$\Phi(\boldsymbol{h}) = \boldsymbol{h} + \sum_{k=1}^{r} w_k(\boldsymbol{h}) \boldsymbol{r}_k (\boldsymbol{a}_k^\top \boldsymbol{h} + b_k - \boldsymbol{r}_k^\top \boldsymbol{h}), \tag{2}$$

where $\boldsymbol{r}_1, \ldots, \boldsymbol{r}_r$ are $r$ $d$-dimensional orthogonal bases, $\boldsymbol{a}_1, \ldots, \boldsymbol{r}_r$ and $b_1, \ldots, b_r$ are $r$ arbitrary vectors and scalars, respectively. Finally, $w_k(\boldsymbol{h}) \in [0, 1]$ are $r$ learnable weights. Put together, $w_k(\boldsymbol{h})(\boldsymbol{a}_k^\top \boldsymbol{h} + b_k - \boldsymbol{r}_k^\top \boldsymbol{h})$ predicts the magnitude of update along direction of basis $\boldsymbol{r}_k$, and BaFT combines $r$ total updates to form the final intervention. Fig 4 illustrates the overall flow of BaFT.

While appears distinct, Lem 2.4 shows that BaFT generalizes ReFT. See its proof in App B.3.

**Lemma 2.4.** *Let* $\mathbf{R} = [\boldsymbol{r}_1; \ldots; \boldsymbol{r}_r] \in \mathbb{R}^{r \times d}$, $\mathbf{A} = [\boldsymbol{a}_1, \ldots, \boldsymbol{a}_r] \in \mathbb{R}^{r \times d}$, $\boldsymbol{b}^\top = (b_1, \ldots, b_k)$, *and* $\mathbf{W}(\boldsymbol{h}) = diag(w_1(\boldsymbol{h}), \ldots, w_r(\boldsymbol{h}))$ *be a diagonal matrix. BaFT in Eqn* (2) *can be expressed as*

$$\Phi(\boldsymbol{h}) = \boldsymbol{h} + \mathbf{R}^\top \mathbf{W}(\boldsymbol{h}) \left( \mathbf{A}\boldsymbol{h} + \boldsymbol{b} - \mathbf{R}\boldsymbol{h} \right). \tag{3}$$

*When using constant weighting* $\mathbf{W}(\boldsymbol{h}) = \mathbf{I}$, *BaFT reduces to ReFT.*

## 2.4 TRAINING OBJECTIVE OF BAFT

We end this section by detailing the training of BaFT. For consistency we use $\phi_l$ to denote the collection of learnable parameters at layer $l$: $\mathbf{R}, \mathbf{A}, \boldsymbol{b}$, and newly introduced parameters in $\mathbf{W}$. Given a set of pre-specified layers $C_l$ that need interventions, we optimize the collection of all learnable parameters $\phi = \{\phi_l\}_{l \in C_l}$ using the following losses.

**Teacher-forcing Loss.** Following Wu et al. (2024), we train BaFT with a language modeling objective, and minimize the cross-entropy loss with teacher-forcing (Lamb et al., 2016) at output positions

$$L_1(\phi) \triangleq - \sum_{i=1}^m \log p_\theta(y_i \mid \boldsymbol{x}\boldsymbol{y}_{<i}; \phi),$$

where the intervention is applied to the last $P$ positions in $\boldsymbol{x}$, together with all entries in $\boldsymbol{y}$.

**Incremental Load Balancing Loss.** When editing multiple pieces of knowledge, different bases need, on average, balanced weights. Otherwise, using a few fixed bases for *all* edits is equivalent to using a *fixed* subspace spanned by these bases, and BaFT will reduce to a smaller ReFT. To avoid this reduction, inspired by the sparse mixture of expert (Shazeer et al., 2017; Fedus et al., 2022), we regularize the *squared coefficient of variation* of $(w_1(\boldsymbol{h}), \ldots, w_r(\boldsymbol{h}))$. However, as new knowledge may emerges one by one, making direct average over multiple samples infeasible, we compute the metric in an incremental way. Namely, when editing the $t$-th knowledge, we minimize

$$\mathcal{R}_{\text{bal}}(\phi) \triangleq \sum_{k=1}^r \frac{(\bar{w}_k(t) - \bar{w}(t))^2}{(r-1)\bar{w}(t)}, \quad \text{where } \bar{w}(t) = \frac{1}{r} \sum_{k=1}^r \bar{w}_k(t),$$

and $\bar{w}_k(t)$ averages weights $w_k$ over the *current* and *past* training samples at selected positions. For incremental optimization, we only minimize $\mathcal{R}_{\text{bal}}(\phi)$ with respect to the current weight on the $t$-th knowledge, as highlighted by expressing $\bar{w}_k(t)$ as a function of current step $t$.

**Locality Regularization.** In some scenarios, it is feasible to obtain examples of irrelevant knowledge during training (Wang et al., 2024d; Yu et al., 2024). Such information can benefit the training of BaFT as well. Following Wang et al. (2024d), we incorporate the margin loss as a regularizer. Let $\boldsymbol{h}$ and $\boldsymbol{h}_{\text{ir}}$ denote the representations of *editing* and *irrelevant* knowledge, respectively, we minimize

$$\mathcal{R}_{\text{loc}}(\phi) = \underbrace{\max(0, \mathbf{W}(\boldsymbol{h}_{\text{ir}}) - \alpha)}_{\text{irr.. weight } w(\boldsymbol{h}_{\text{ir}}) \le \alpha} + \underbrace{\max(0, \beta - \mathbf{W}(\boldsymbol{h}))}_{\text{edit. weight } w(\boldsymbol{h})t \ge \beta} + \underbrace{\max(0, \gamma - (\mathbf{W}(\boldsymbol{h})_{\max} - \mathbf{W}(\boldsymbol{h}_{\text{ir}})_{\max})}_{\text{edit weight } \ge \text{loc weight}}.$$

At a colloquial level, $\mathcal{R}_{\text{loc}}(\phi)$ encourages that weights for irrelevant knowledge should be as small as $\alpha$, editing knowledge's weight should be no less as $\beta$, and at the same time, the most important weights from the two groups should have a gap that is as large as $\gamma$.

In execution, we rescale the three terms to the same magnitude and solve the following objective

$$\min_\phi L(\phi) \triangleq \min_\phi L_1(\phi) + \mathcal{R}_{\text{bal}}(\phi) + \mathcal{R}_{\text{loc}}(\phi). \tag{4}$$

ReFT, as a special case of BaFT, only minimizes $L_1(\phi)$.

## 3 EXPERIMENT

We test the proposed BaFT for knowledge editing on three 7B-level autoregressive language models (LMs) over five public benchmarks. Ablation studies are also conducted. Experiment results show that BaFT can achieve excellent performance at much better parameter efficiency.

### 3.1 EXPERIMENT SETUP

**Base Models.** We conduct experiments on three representative LLMs from different model families. **LLaMA 2-7b** (and **LLaMA 2-7b-Chat**) (Touvron et al., 2023) have been widely studied in the literature (Zhang et al., 2024b; Wang et al., 2024d) and we follow this convention. Trending **LLaMA 3-8b-Instruct** (Dubey et al., 2024) and **Gemma 1.1-7b-Instruct** (Team et al., 2024) are also studied. From now on, we refer to the three LLMs as **LLaMA 2(-chat), LLaMA 3,** and **Gemma** for brevity.

**Tasks.** Following previous works (Wang et al., 2023; Zhang et al., 2024b), we edit different kinds of knowledge: WikiData$_{recent}$, WikiData$_{counterfact}$ (Cohen et al., 2024), WikiBio (Hartvigsen et al., 2024), ConvSent (Mitchell et al., 2022), and ZsRE (Yao et al., 2023). Due to page limitation, we refer readers to Zhang et al. (2024b) for more benchmark details. When editing an LLM, three scenarios are considered. **Single Editing** updates one piece of knowledge at a time. **Continual Editing** and **Batched Editing**, on the other hand, update multiple pieces of knowledge in a sequential or batched way. The two latter are more challenging due to potential forgetting and knowledge conflicting problems, as observed in the literature (Hartvigsen et al., 2024; Wang et al., 2024d).

**Baselines.** We follow Zhang et al. (2024b); Wang et al. (2024e) and choose AdaLoRA (Zhang et al., 2023), ROME and FT-L (Meng et al., 2022a), and MEMIT (Meng et al., 2022b) as baselines. In continual editing scenarios, we further include representative memory-based methods GRACE (Hartvigsen et al., 2024), MELO (Yu et al., 2024), and WISE (Wang et al., 2024d). All these baselines, same as ours, *do not require a larges-scale hard-to-access training data, or training additional models*: AdaLoRA learns a low-rank update for model parameters on the new knowledge while keeping less important parameters unchanged, thereby achieving a highly efficient and precise PEFT. ROME applies a causal-tracing analysis to identify the layer wherein the knowledge is stored and then solves an analytic rank-one update. FT-L, on the other hand, directly finetunes the layer identified by ROME with an additional KL divergence loss. MEMIT extends ROME to a batched editing setting by identifying a series of layers to edit and finding the updates as least squares solutions. GRACE, MELO, and WISE are specialized for continual editing. They leverage side parameters to save new knowledge and learn gating mechanism to determine whether pre-trained or new knowledge should be used during inference. Finally, we include ReFT as a baseline that uses a subspace of the same rank as BaFT.

**Evaluation Criteria.** We evaluate the performance from multiple aspects (Zhang et al., 2024b; Wang et al., 2024d). Given an edited model, **reliability (Rel.)** evaluates whether it successfully learns the new knowledge; **generality (Gen.)** measures to what extent it can generalize to rephrased knowledge inquiries; **locality (Loc.)** quantifies how much the model can retain its original output on irrelevant knowledge inquiries; **portability (Por.)** checks if the model is able to transfer new knowledge to related content. We report the average of different metrics[1] for more complete comparisons.

**Implementation Details.** Our experiments are conducted with EasyEdit (Wang et al., 2024e). More implementation details and hyper-parameters can be found in App C.

### 3.2 SINGLE EDITING PERFORMANCE

We evaluate the effectiveness of the proposed BaFT for conducting Single Editing on WikiData$_{recent}$, WikiData$_{counterfact}$, WikiBio, and ConvSent (only supports LLaMA family). The four benchmarks do not contain irrelevant data. Consequently, *BaFT training does not involve the locality regularization*.

Single Editing results are reported in Tab 1. The proposed BaFT performs highly competitively in all cases. BaFT and ReFT use a subspace of the same rank to edit representations, so an ideal BaFT should achieve reliability comparable to ReFT that can edit representations freely. Indeed, BaFT maintains a better editing-locality trade-off: it consistently achieves better locality and portability

---

[1]Not all benchmarks support all metrics.

than ReFT with no degradation of reliability. In comparison, other baselines suffer from notable editing-locality trade-off, i.e., achieve high reliability at a price of low locality. These methods also exhibit significant performance gaps when editing different LLMs. These results demonstrates BaFT as a new promising editing solution.

Table 1: Single Editing performance on four benchmark datasets. Results marked with "♡" are taken from Zhang et al. (2024b). Unsupported experiments are marked with "✗". Best Avg. results are in **bold** and second best are underlined.

| | Wiki$_{recent}$ | | | | Wiki$_{counterfact}$ | | | | WikiBio | | | ConvSent |
|---|---|---|---|---|---|---|---|---|---|---|---|---|
| | | | | | LLaMA 2-7b-chat | | | | | | | |
| | Rel. | Por. | Loc. | Avg. | Rel. | Por. | Loc. | Avg. | Rel. | Loc. | Avg. | Rel. |
| AdaLoRA♡ | 1.00 | 0.65 | 0.56 | 0.74 | 1.00 | 0.70 | 0.70 | 0.80 | 1.00 | 0.81 | 0.91 | 0.45 |
| FT-L♡ | 0.56 | 0.41 | 0.44 | 0.47 | 0.45 | 0.34 | 0.50 | 0.51 | 0.66 | 0.80 | 0.73 | 0.50 |
| ROME♡ | 0.97 | 0.55 | 0.55 | 0.69 | 0.99 | 0.56 | 0.52 | 0.69 | 0.96 | 0.63 | 0.80 | 0.46 |
| MEMIT♡ | 0.97 | 0.56 | 0.52 | 0.68 | 0.98 | 0.59 | 0.47 | 0.68 | 0.94 | 0.62 | 0.78 | 0.45 |
| ReFT | 1.00 | 0.60 | 0.71 | 0.77 | 1.00 | 0.72 | 0.78 | 0.83 | 1.00 | 0.91 | 0.96 | 1.00 |
| BaFT (Ours) | 1.00 | 0.61 | 0.73 | **0.78** | 1.00 | 0.72 | 0.81 | **0.84** | 1.00 | 0.94 | **0.97** | 1.00 |
| | | | | | LLaMA 3-8b-Instruct | | | | | | | |
| | Rel. | Por. | Loc. | Avg. | Rel. | Por. | Loc. | Avg. | Rel. | Loc. | Avg. | Rel. |
| AdaLoRA | 1.00 | 0.61 | 0.45 | 0.69 | 1.00 | 0.74 | 0.51 | 0.75 | 1.00 | 0.79 | 0.90 | 1.00 |
| FT-L | 0.47 | 0.27 | 0.22 | 0.32 | 0.43 | 0.32 | 0.22 | 0.32 | 0.56 | 0.71 | 0.64 | 0.52 |
| ROME | 0.99 | 0.58 | 0.49 | 0.69 | 0.99 | 0.58 | 0.41 | 0.66 | 0.92 | 0.68 | 0.80 | 0.98 |
| MEMIT | 0.99 | 0.54 | 0.48 | 0.67 | 0.99 | 0.58 | 0.43 | 0.67 | 0.96 | 0.71 | 0.84 | 0.32 |
| ReFT | 1.00 | 0.62 | 0.62 | **0.75** | 1.00 | 0.72 | 0.74 | **0.82** | 1.00 | 0.87 | 0.94 | 0.98 |
| BaFT (Ours) | 1.00 | 0.62 | 0.64 | **0.75** | 1.00 | 0.72 | 0.75 | **0.82** | 1.00 | 0.91 | **0.96** | 0.96 |
| | | | | | Gemma 1.1-7b-Instruct | | | | | | | |
| | Rel. | Por. | Loc. | Avg. | Rel. | Por. | Loc. | Avg. | Rel. | Loc. | Avg. | Rel. |
| AdaLoRA | 1.00 | 0.58 | 0.28 | 0.62 | 1.00 | 0.70 | 0.35 | 0.68 | 1.00 | 0.70 | 0.85 | ✗ |
| FT-L | 0.35 | 0.20 | 0.03 | 0.26 | 0.20 | 0.18 | 0.01 | 0.13 | 0.24 | 0.14 | 0.19 | ✗ |
| ROME | 0.79 | 0.38 | 0.27 | 0.48 | 0.82 | 0.47 | 0.27 | 0.52 | 0.47 | 0.31 | 0.39 | ✗ |
| ReFT | 1.00 | 0.54 | 0.55 | 0.70 | 1.00 | 0.63 | 0.72 | 0.78 | 1.00 | 0.82 | 0.91 | ✗ |
| BaFT (Ours) | 1.00 | 0.54 | 0.58 | **0.71** | 1.00 | 0.62 | 0.77 | **0.80** | 1.00 | 0.85 | **0.93** | ✗ |

## 3.3 CONTINUAL AND BATCHED EDITING PERFORMANCE.

Next, we study the two challenging scenarios, where massive editings are conducted in a sequential (continual) or batched way. We follow Wang et al. (2024d) and experiment with LLaMA 2 (non-chat version), LLaMA 3, and Gemma on ZsRE. We note that the state-of-the-art continual editing method WISE contains substantially larger parameter size and is much more computationally expensive. For fair comparison, we include WISE$_{light}$, a lightweight version of WISE that contains 1/8 learnable parameters of the original WISE to make its training affordable. We want to highlight that WISE$_{light}$ does not change editing mechanism[2], and still contains more learnable parameters than BaFT and ReFT (10 and 20 times respectively). Learnable parameters used in different methods, along with their time consumptions, are reported in Tab 3.

**Continual Editing Performance.** Tab 2 presents the main results of continually editing 1000 pieces of ZsRE knowledge. BaFT again achieves remarkable editing performance while maintaining excellent locality on LLMs from different families, reaching the best two in nearly all scenarios. In comparison, standard methods AdaLoRA, FT-L, ROME, and MEMIT encounter considerable performance gaps over different LLMs. Meanwhile, they fall short in editing multiple pieces of knowledge that emerge sequentially. WISE performs slightly better but its parameter efficiency is much lower, as we will show soon. GRACE is designed for continual editing but still suffers from failure on editing Gemma. These methods might benefit from a more extensive hyper-parameter tuning for each LLM. Nonetheless, their prolonged running time makes this process expensive, if not unaffordable.

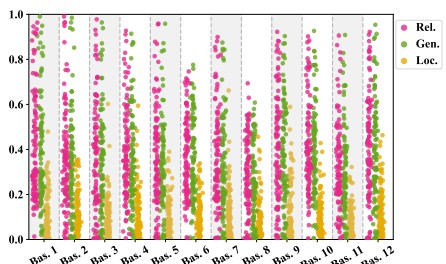

Figure 2: Bases weights used for editing and irrelevant knowledge (averaged over different positions).

---

[2]WISE finds an important FFN layer to conduct knowledge editing. For each new knowledge, it finetunes a small portion of *randomly* selected parameters in this layer. WISE$_{light}$ uses a smaller randomly chosen pool.

When comparing BaFT and ReFT with each other, we note that as in Single Editing, BaFT maintains, if not surpasses, the editing ability of ReFT. In addition, when the editing number $T$ increases, BaFT shows excellent robustness against forgetting, as indicated by its capability of preserving high locality in all scenarios. We further visualize bases weights in Fig 2, where a one-layer BaFT is used to edit LLaMA 2 on 100 ZsRE knowledge with $T = 10$ (achieved reliability, generality, and locality are 0.75, 0.71, and 0.98 respectively). Rel., Gen., and Loc. refers to *new*, *rephrased*, and *unrelated* knowledge, respectively. We note that BaFT evenly distributes the editing over all bases, and unrelated knowledge receives significantly lower weights. These results confirm that BaFT leverages the fine-grained basis-level control as designed in Sec 2, thereby excelling at Continual Editing.

Table 2: Continual Editing performance on ZsRE dataset, evaluated after conducting $T$ times of editing sequentially. Results marked with "♡" are taken from Wang et al. (2024d). Best Avg. results are in **bold** and second best are underlined.

| | $T = 1$ | | | | $T = 10$ | | | | $T = 100$ | | | | $T = 1000$ | | | |
|---|---|---|---|---|---|---|---|---|---|---|---|---|---|---|---|---|
| | Rel. | Gen. | Loc. | Avg. | Rel. | Gen. | Loc. | Avg. | Rel. | Gen. | Loc. | Avg. | Rel. | Gen. | Loc. | Avg. |
| **LLaMA 2-7b** | | | | | | | | | | | | | | | | |
| AdaLoRA | 1.00 | 0.90 | 0.92 | 0.94 | 0.39 | 0.38 | 0.50 | 0.42 | 0.06 | 0.06 | 0.06 | 0.06 | 0.00 | 0.00 | 0.00 | 0.00 |
| FT-L | 0.57 | 0.53 | 0.96 | 0.69 | 0.35 | 0.31 | 0.12 | 0.26 | 0.29 | 0.26 | 0.09 | 0.21 | 0.24 | 0.20 | 0.25 | 0.23 |
| ROME | 0.96 | 0.91 | 0.98 | 0.95 | 0.80 | 0.76 | 0.77 | 0.78 | 0.18 | 0.18 | 0.07 | 0.12 | 0.00 | 0.01 | 0.00 | 0.00 |
| MEMIT | 0.95 | 0.90 | 0.99 | 0.95 | 0.77 | 0.74 | 0.90 | 0.80 | 0.25 | 0.24 | 0.19 | 0.02 | 0.04 | 0.04 | 0.02 | 0.03 |
| MELO | 1.00 | 0.40 | 0.99 | 0.80 | 0.95 | 0.40 | 0.99 | 0.78 | 0.61 | 0.40 | 0.99 | 0.67 | 0.40 | 0.40 | 0.99 | 0.60 |
| GRACE♡ | 0.98 | 0.08 | 1.00 | 0.69 | 0.96 | 0.00 | 1.00 | 0.65 | 0.96 | 0.00 | 1.00 | 0.65 | 0.97 | 0.08 | 1.00 | 0.68 |
| WISE$_{full}$♡ | 0.98 | 0.92 | 1.00 | **0.97** | 0.94 | 0.88 | 1.00 | **0.94** | 0.90 | 0.81 | 1.00 | **0.90** | 0.77 | 0.72 | 1.00 | **0.83** |
| WISE$_{light}$ | 0.95 | 0.83 | 1.00 | 0.93 | 0.93 | 0.74 | 1.00 | 0.89 | 0.83 | 0.73 | 0.99 | 0.85 | 0.49 | 0.47 | 1.00 | 0.65 |
| ReFT | 1.00 | 0.95 | 0.94 | 0.96 | 0.90 | 0.85 | 0.88 | 0.87 | 0.78 | 0.74 | 0.83 | 0.78 | 0.58 | 0.56 | 0.73 | 0.62 |
| BaFT (Ours) | 1.00 | 0.94 | 0.97 | **0.97** | 0.89 | 0.84 | 0.97 | 0.90 | 0.75 | 0.70 | 0.98 | 0.81 | 0.63 | 0.60 | 0.98 | 0.74 |
| **LLaMA 3-8b-Instruct** | | | | | | | | | | | | | | | | |
| | Rel. | Gen. | Loc. | Avg. | Rel. | Gen. | Loc. | Avg. | Rel. | Gen. | Loc. | Avg. | Rel. | Gen. | Loc. | Avg. |
| AdaLoRA | 1.00 | 0.99 | 0.85 | 0.95 | 0.27 | 0.26 | 0.26 | 0.26 | 0.03 | 0.03 | 0.01 | 0.02 | 0.00 | 0.00 | 0.00 | 0.00 |
| FT-L | 0.51 | 0.52 | 0.68 | 0.57 | 0.25 | 0.20 | 0.03 | 0.16 | 0.19 | 0.16 | 0.02 | 0.12 | 0.16 | 0.14 | 0.01 | 0.10 |
| ROME | 0.99 | 0.96 | 0.96 | 0.97 | 0.62 | 0.63 | 0.42 | 0.56 | 0.07 | 0.07 | 0.01 | 0.05 | 0.03 | 0.03 | 0.00 | 0.02 |
| MEMIT | 0.99 | 0.96 | 0.98 | **0.98** | 0.68 | 0.66 | 0.71 | 0.68 | 0.03 | 0.03 | 0.02 | 0.03 | 0.00 | 0.00 | 0.00 | 0.00 |
| MELO | 1.00 | 0.29 | 1.00 | 0.76 | 0.97 | 0.30 | 1.00 | 0.76 | 0.55 | 0.31 | 0.99 | 0.62 | 0.31 | 0.30 | 0.99 | 0.53 |
| GRACE | 0.33 | 0.00 | 0.54 | 0.29 | 0.33 | 0.02 | 0.56 | 0.30 | 0.33 | 0.02 | 0.57 | 0.31 | 0.33 | 0.02 | 0.55 | 0.30 |
| WISE$_{light}$ | 0.95 | 0.91 | 0.99 | 0.95 | 0.82 | 0.76 | 1.00 | 0.86 | 0.63 | 0.57 | 1.00 | 0.73 | 0.39 | 0.37 | 1.00 | 0.59 |
| ReFT | 1.00 | 0.97 | 0.93 | 0.97 | 0.90 | 0.84 | 0.87 | 0.87 | 0.68 | 0.61 | 0.74 | 0.68 | 0.48 | 0.45 | 0.64 | 0.52 |
| BaFT (Ours) | 1.00 | 0.95 | 0.96 | 0.97 | 0.89 | 0.82 | 0.95 | **0.89** | 0.70 | 0.64 | 0.93 | **0.76** | 0.50 | 0.49 | 0.93 | **0.64** |
| **Gemma 1.1-7b-Instruct** | | | | | | | | | | | | | | | | |
| | Rel. | Gen. | Loc. | Avg. | Rel. | Gen. | Loc. | Avg. | Rel. | Gen. | Loc. | Avg. | Rel. | Gen. | Loc. | Avg. |
| AdaLoRA | 1.00 | 0.97 | 0.67 | 0.88 | 0.19 | 0.20 | 0.18 | 0.19 | 0.03 | 0.03 | 0.01 | 0.02 | 0.00 | 0.00 | 0.00 | 0.00 |
| FT-L | 0.28 | 0.33 | 0.09 | 0.23 | 0.14 | 0.06 | 0.00 | 0.07 | 0.07 | 0.04 | 0.00 | 0.04 | 0.05 | 0.04 | 0.00 | 0.03 |
| ROME | 0.75 | 0.71 | 0.88 | 0.78 | 0.18 | 0.18 | 0.05 | 0.14 | 0.01 | 0.01 | 0.01 | 0.01 | 0.00 | 0.00 | 0.00 | 0.00 |
| MELO | 1.00 | 0.20 | 1.00 | 0.73 | 0.96 | 0.23 | 1.00 | 0.73 | 0.52 | 0.26 | 0.95 | 0.58 | 0.26 | 0.25 | 0.95 | 0.49 |
| GRACE | 0.39 | 0.00 | 1.00 | 0.46 | 0.39 | 0.01 | 1.00 | 0.47 | 0.39 | 0.01 | 1.00 | 0.47 | 0.39 | 0.01 | 1.00 | 0.47 |
| WISE$_{light}$ | 0.99 | 0.96 | 1.00 | **0.98** | 0.90 | 0.84 | 0.99 | **0.91** | 0.79 | 0.71 | 0.95 | **0.82** | 0.48 | 0.42 | 0.98 | **0.63** |
| ReFT | 1.00 | 0.86 | 0.91 | 0.92 | 0.92 | 0.81 | 0.81 | 0.85 | 0.66 | 0.58 | 0.69 | 0.64 | 0.50 | 0.46 | 0.65 | 0.54 |
| BaFT (Ours) | 1.00 | 0.84 | 0.94 | 0.93 | 0.92 | 0.80 | 0.92 | 0.88 | 0.70 | 0.62 | 0.92 | 0.75 | 0.48 | 0.45 | 0.92 | 0.62 |

Table 3: Parameter size and editing time with an NVIDIA V100 32-GB GPU (averaged over 100 samples). ROME, MEMIT, and GRACE do not contain pre-specified learnable parameters.

| | LLaMA 2-7b(-chat) | | LLaMA 3-8b-Instruct | | Gemma 1.1-7b-Instruct | |
|---|---|---|---|---|---|---|
| | # Params. | Time (sec./edit) | # Params. | Time (sec./edit) | # Params. | Time (sec./edit) |
| AdaLoRA | 6,292,224 | 26.24 | 5,112,576 | 28.71 | 4,817,568 | 44.24 |
| FT-L | 45,088,768 | 9.73 | 58,720,256 | 10.84 | 75,497,472 | 11.95 |
| ROME | / | 27.27 | / | 25.01 | / | 52.07 |
| MEMIT | / | 20.01 | / | 25.35 | / | / |
| GRACE | / | 34.38 | / | 87.08 | / | 43.45 |
| WISE$_{light}$ | 5,636,096 | 58.00 | 7,340,032 | 65.77 | 9,437,184 | 20.20 |
| ReFT | 393,264 | 10.99 | 393,264 | 9.33 | 294,960 | 7.79 |
| BaFT (Ours) | 606,256 | 13.46 | 606,256 | 12.69 | 454,704 | 10.13 |

**Batched Editing Performance.** We further compare BaFT and ReFT against baselines that *admit* batched data for editing, namely, AdaLoRA, FT-L, and MEMIT. LMs were edited on ZsRE dataset, and batch sizes were set to 10 and 50 respectively.

We visualize the average of reliability, generality, and locality in Fig 3, and defer the complete results to App D. The proposed BaFT again achieved a great balance between good edit success and high locality, outperforming all baselines in 5 out of 6 scenarios. Surprisingly, when $T = 10$, LoRA

and MEMIT were capable of benefiting from editing multiple samples in a batch than one by one. We conjecture that learning multiple pieces of knowledge in a batch helps mitigate their overfitting on any single knowledge, thereby weakened the forgetting problem to some extent. This finding suggests that caching more knowledge and editing them in a batch can be beneficial in some cases.

**Parameter Efficiency.** Continual and Batched Editing involve learning more knowledge than in Single Editing. As a result, achieving good editing performance while maintaining high parameter efficiency is non-trivial, as using fewer parameters increases the workload of each parameter to *learn* more knowledge. We note that while WISE$_{\text{light}}$ achieved comparable performance to BaFT, its parameter efficiency was much lower: on LLaMA 2-7b, the edit success dropped from 0.77 (of WISE) to 0.49 when editing 1000 pieces of knowledge, around 22% lower than BaFT which uses 10 times less parameters, as per Table 3. Similar trends can be found when making comparison with LoRA in Batched Editing scenarios. In conclusion, BaFT is capable of achieving much better parameter efficiency than existing methods.

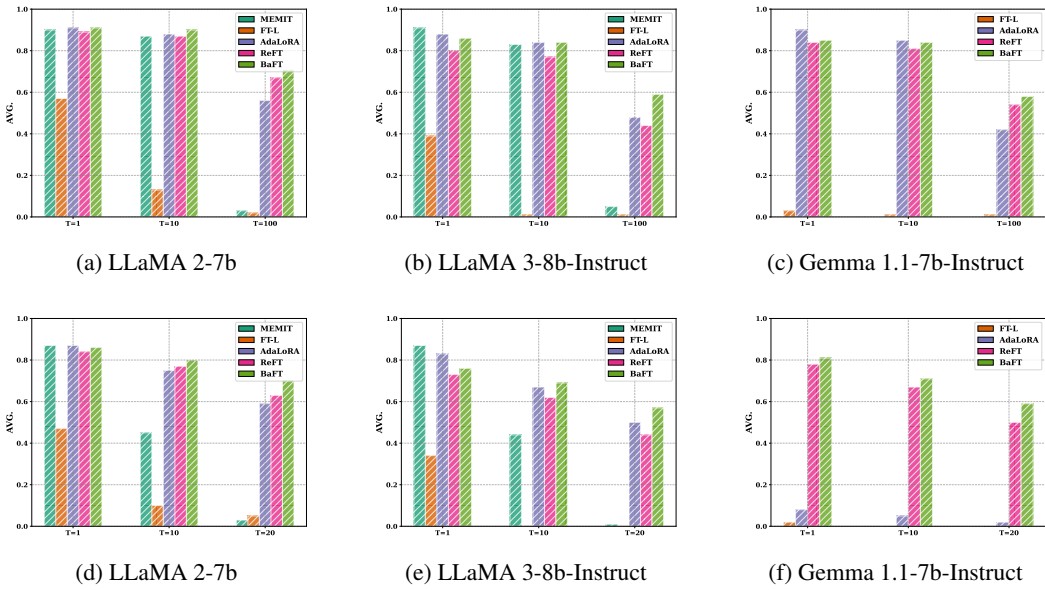

Figure 3: Batched Editing Performance under sequence length. The first row uses batch size 10 and the second row uses batch size 50.

### 3.4 ABLATION STUDY

We end this section with an ablation study on BaFT to showcase how each component contributes to the final performance. Results from continually editing LLaMA 2-7b with 100 ZsRE knowledge are presented in Tab 4. We note that introducing a coarse-grained *subspace-level* weighting (*ss-w*) which assigns all bases with the same weight along did not benefit ReFT. Moreover, both locality regularization (*lr*) and fine-grained basis-level weighting (*ba-w*) helped improve locality. Remarkably, the basis-level weighting, as observed in all Single Editing scenarios, did not lead to edit performance degradation. Locality regularization, while greatly improved the locality, induced a trade-off with editing performance at the same time. Notably, the degradation is amplified when the subspace-level weighting was used, echoing well with our theoretical analysis.

In conclusion, the proposed BaFT makes two improvements over ReFT. First, the weighting offers a fine-grained level learning, leading to better locality without hurting editing performance. Second, a fine-grained basis-level control allows one to regularize locality by altering only the important parts, leading to a better empirical editing-locality trade-off.

### 4 RELATED WORKS

Existing editing methods mainly fall into two classes.

**Internal Storage** updates model parameters for the adaptation. Early efforts involved fine-tuning a LLM directly but suffered from severe forgetting of original knowledge (Wang et al., 2023). For more precise editing, Zhu et al. (2020) imposed a relaxed $\ell_2$ norm constraint on the parameter updates, and Huang et al. (2023); Dong et al. (2022a) limited the updates to some specific feed-forward network (FFN) layers, based on findings that knowledge is often stored therein (Dai et al., 2021). For further refinement, the *locate-and-edit*

Table 4: Component effects in BaFT.

|  | Rel. | Gen. | Loc. | Avg. |
|---|---|---|---|---|
| ReFT | 0.76 | 0.71 | 0.84 | 0.77 |
| *+ss-w.* | 0.74 | 0.68 | 0.81 | 0.74 |
| *+ba-w* | 0.77 | 0.71 | 0.86 | 0.78 |
| *+ss-w&lr* | 0.67 | 0.61 | 0.99 | 0.76 |
| BaFT | 0.73 | 0.67 | 0.98 | 0.79 |

paradigm (Meng et al., 2022a;b) first identifies the layer storing a specific knowledge, and then modifies its parameters in an analytic form or via least squared solution. On the other hand, PEFT methods such as AdaLoRA (Zhang et al., 2023) also provided performance on par with locating-based solutions (Wu et al., 2023; Wang et al., 2024b). However, these methods are *parameter-based* and offer a similar level of control, in the sense that all inputs are altered in the same way. As a result, they suffer from an equal level of *editing-locality* trade-off (Wang et al., 2023; 2024d). These findings raised a question as *to what extent knowledge can be accurately attributed to some specific parameters* (Hase et al., 2024). Inspired by the recent advance of improving a LLM's general ability such as commonsense reasoning by fine-tuning its representations (Wu et al., 2024), in this work we show that updating representations at only a few locations can provide strong editing performance. By pushing the fine-tuning towards a new basis-level, our BaFT achieved better fine-grained control and superior editing-locality trade-off.

**External Storage** resorts to external memories without changing original parameters. Methods include meta-learning based MEND (Mitchell et al., 2021) and its multi-task version InstructEdit (Zhang et al., 2024a), IKE (Zheng et al., 2023) and LTE (Jiang et al., 2024) that bear the similarity to Retrieval-Augmented Generation (Gao et al., 2023; Wang et al., 2024a; Xu et al., 2024; Yu et al., 2025; Liu et al., 2025b), augmentation based StableKE (Wei et al., 2024), and proxy model based SERAC (Mitchell et al., 2022). Notwithstanding, these methods need large-scaled *hard-to-access* data to retrieve from (e.g., IKE, LTE), or to train extra model on (e.g., MEND, InstructEdit, SERAC). As a consequence, they have limited practicality and fall short on Continual Editing that requires frequent updates (Wang et al., 2024d). Recently, methods specialized for Continual Editing were proposed (Hartvigsen et al., 2024; Yu et al., 2024; Wang et al., 2024d). These approaches injected lightweight adapters (Hartvigsen et al., 2024) or weight copies (Wang et al., 2024d) to memorize new knowledge, and learned some gating mechanism to determine whether original or new knowledge to use. Specifically, GRACE (Hartvigsen et al., 2024) maintained a code-book to determine which adapter will be used based on representation similarity, and MELO (Yu et al., 2024) used dynamic LoRA. WISE (Wang et al., 2024d) learned activation threshold to trigger new learned weights. However, these methods have several limitations. First, they often show unsatisfactory generalizability, as observed in Wang et al. (2024d) and confirmed in our experiments. Second, they require prolonged training (and inference) time, due to the need of maintaining non-constant numbers of external memories. Finally, existing gating mechanisms cannot be learned when multiple pieces of knowledge appear, making them incompatible for Batched Editing. In comparison, BaFT learns a pre-specified set of parameters and lets bases weights play the role of gating. This design makes BaFT suitable for both Continual and Batched Editing. Moreover, as *editing* and *activation* are conducted in a representation subspace, BaFT is able to achieve good generalizability at better parameter efficiency.

## 5 CONCLUSION AND FUTURE WORKS

In this work, we propose a new representation based method towards more efficient knowledge editing. Grounded in a theoretical analysis, we show that updating all selected representations with one linear subspace in a predetermined manner imposes a tension in editing-locality trade-off. Subsequently, BaFT as a better solution is proposed. Given a representation, BaFT first computes a weight for each basis that spans the linear subspace, then conducts a linear update along this basis direction. Because bases weights are determined from the current representation with non-linear functions, BaFT fine-tunes the representation in a non-linear way. This fine-grained control leads to better performance on editing three representative LLMs in various scenarios, on par with or outperforming the strongest baselines at much better parameter efficiency. As detailed in App A, there are some limitations in this work, and we plan to work on in our future work.

## ACKNOWLEDGEMENT

This work is supported in part by the US National Science Foundation under grant NSF IIS-2141037. Any opinions, findings, and conclusions or recommendations expressed in this material are those of the author(s) and do not necessarily reflect the views of the National Science Foundation.

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

## A MORE DISCUSSIONS AND LIMITATIONS ON BAFT

In this section we provide more discussions on the proposed BaFT. Fig 4 demonstrates the workflow of our method. There are also some limitations in this work, and we plan to explore in the future.

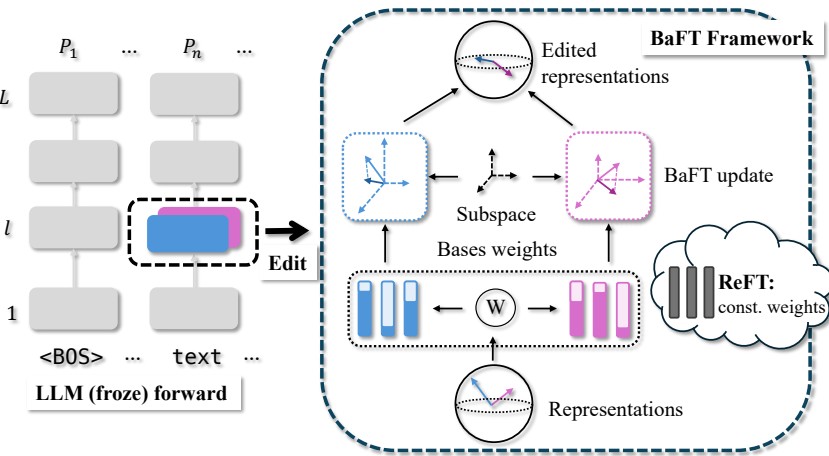

Figure 4: BaFT learns basis-level weights to edit different representations (highlighted in different colors). When using constant weights, BaFT reduces to ReFT.

First, The empirical success of BaFT was mainly established on standard benchmarks EasyEdit (Wang et al., 2024e), which may not be sufficient to reflect the diverse real-world applications. Second, BaFT as a generalization of ReFT requires hyper-parameter tuning to determine proper positions and layers to add interventions. Our choice was selected based on recommended values from ReFT (Wu et al., 2024). We plan to explore automating this process by imposing proper sparsity constraints on weights in our future work. Third, the promising performance of BaFT demonstrates its potential for efficient knowledge editing. However, it is still an open question if representation-based method is capable of fitting any editing (or updates) learnable for parameter-based methods. In other words, it is unknown if there is some knowledge that can be learned by parameter-based method but is unlearnable by updating representations. We plan to explore this direction in our future work.

## B OMITTED PROOF

We include omitted proof here.

### B.1 PROOF OF THM 2.3

We start with restating the two assumptions and the theorem.

**Assumption B.1.** Let text $x$ encodes $s, r$, text $y$ generated by the LM will convey $o$ if its intermediate representation takes some targeted value $t$.

**Assumption B.2.** For any $h$ carrying some high-level knowledge, there exists a positive $\varepsilon(h)$-radius $\ell_2$ ball $B(h, \varepsilon(h))$ around $h$ such that any $h' \in B(h, \varepsilon(h))$ conveys the same knowledge, we refer to $B(h, \varepsilon(h))$ as a *stable-ball* of $h$.

**Theorem B.3.** *When finetuning a LM, ReFT learns to update the old representation $h_0$ to targeted $t = \Phi(h_0)$. If ReFT maintains good generality such that $\forall\, h \in B(h_0, \varepsilon(h_0))$,*

$$\|\Phi(h) - \Phi(h_0)\| = \|\Phi(h) - t\| < \varepsilon(t),$$

*where $\|\cdot\|$ denote the $\ell_2$ norm. Then for any irrelevant input $h_{ir}$ with a small stable-ball radius*

$$\varepsilon(h_{ir}) < \frac{\|t - h_0\| - (\varepsilon(t) + \varepsilon(h_0))}{\varepsilon(t) + 2\varepsilon(h_0)}\varepsilon(h_0),$$

*and is not too far from $h_0$ such that*

$$\|h_{ir} - h_0\| = \varepsilon(h_{ir}) + \varepsilon(h_0),$$

*ReFT will output $\Phi(h_{ir}) \notin B(h_{ir}, \varepsilon(h_{ir}))$ and break its locality guarantee.*

*Proof.* First, since old and new knowledge associates with different objects $o$, by Asmp 2.2, $\boldsymbol{h}_0$ and $\boldsymbol{t}$ must have non-overlapped stable-ball. Otherwise, we can find

$$\boldsymbol{h} \in B(\boldsymbol{h}_0, \varepsilon(\boldsymbol{h}_0)) \cap B(\boldsymbol{t}, \varepsilon(\boldsymbol{t})),$$

that preserves the knowledge of both $\boldsymbol{h}_0$ and $\boldsymbol{t}$ that are different, which is impossible. This implies

$$\|\boldsymbol{t} - \boldsymbol{h}_0\| \geq \varepsilon(\boldsymbol{t}) + \varepsilon(\boldsymbol{h}_0).$$

In addition, by definition of ReFT, we have

$$\begin{aligned}
\boldsymbol{t} - \boldsymbol{h}_0 &= \Phi(\boldsymbol{h}_0) - \boldsymbol{h}_0 \\
&= \boldsymbol{h}_0 + \mathbf{R}^\top(\mathbf{A}\boldsymbol{h}_0 + \boldsymbol{b}) - \boldsymbol{h}_0 \\
&= \mathbf{R}^\top(\mathbf{A} - \mathbf{R})\boldsymbol{h}_0 + \mathbf{R}^\top\boldsymbol{b} \\
&\overset{(a)}{=} \mathbf{H}\boldsymbol{h}_0 + \mathbf{R}^\top\boldsymbol{b},
\end{aligned}$$

where in the last step $(a)$, we defined $\mathbf{H} \triangleq \mathbf{R}^\top(\mathbf{A} - \mathbf{R})$ for simplicity.

Next, according to the *generality* condition, for any $\boldsymbol{h} \in B(\boldsymbol{h}_0, \varepsilon(\boldsymbol{h}_0))$, we have

$$\begin{aligned}
&\|\Phi(\boldsymbol{h}) - \Phi(\boldsymbol{h}_0)\| \\
=&\|(\mathbf{I} + \mathbf{R}^\top(\mathbf{A} - \mathbf{R}))(\boldsymbol{h} - \boldsymbol{h}_0)\| \\
=&\|(\mathbf{I} + \mathbf{H})(\boldsymbol{h} - \boldsymbol{h}_0)\| \\
<&\varepsilon(\boldsymbol{t}).
\end{aligned}$$

Since $\boldsymbol{h}$ can take any direction, we know $\boldsymbol{h} - \boldsymbol{h}_0$ can be an arbitrary vector that has norm no greater than $\varepsilon(\boldsymbol{h}_0)$. Let $\boldsymbol{h} - \boldsymbol{h}_0$ takes the direction of the first right singular vector, then

$$\|(\mathbf{I} + \mathbf{H})(\boldsymbol{h} - \boldsymbol{h}_0)\| = \sigma_{\max}(\mathbf{I} + \mathbf{H})\|\boldsymbol{h} - \boldsymbol{h}_0\| < \varepsilon(\boldsymbol{t}).$$

This implies that the operator norm of $\mathbf{I} + \mathbf{H}$, denoted by $\sigma_{\max}$, is upper bounded by

$$\sigma_{\max}(\mathbf{I} + \mathbf{H}) \leq \frac{\varepsilon(\boldsymbol{t})}{\varepsilon(\boldsymbol{h}_0)}.$$

By triangle inequality of the operator norm (Belitskii et al., 2013), we further know

$$\sigma_{\max}(\mathbf{H}) = \sigma_{\max}(\mathbf{I} + \mathbf{H} - \mathbf{I}) \leq \sigma_{\max}(\mathbf{I} + \mathbf{H}) + \sigma_{\max}(\mathbf{I}) \leq \frac{\varepsilon(\boldsymbol{t})}{\varepsilon(\boldsymbol{h}_0)} + 1.$$

Now, for any irrelevant $\boldsymbol{h}_{\mathrm{ir}}$, we have

$$\begin{aligned}
&\|\Phi(\boldsymbol{h}_{\mathrm{ir}}) - \boldsymbol{h}_{\mathrm{ir}}\| \\
=&\|\mathbf{H}\boldsymbol{h}_{\mathrm{ir}} + \mathbf{R}^\top\boldsymbol{b}\| \\
\overset{(a)}{=}&\|\mathbf{H}\boldsymbol{h}_{\mathrm{ir}} + (\boldsymbol{t} - \boldsymbol{h}_0) - \mathbf{H}\boldsymbol{h}_0\| \\
=&\|\mathbf{H}(\boldsymbol{h}_{\mathrm{ir}} - \boldsymbol{h}_0) + (\boldsymbol{t} - \boldsymbol{h}_0)\| \\
\overset{(b)}{\geq}&\big|\|(\boldsymbol{t} - \boldsymbol{h}_0)\| - \|\mathbf{H}(\boldsymbol{h}_{\mathrm{ir}} - \boldsymbol{h}_0)\|\big|,
\end{aligned} \tag{†}$$

where $(a)$ substitutes

$$\boldsymbol{t} - \boldsymbol{h}_0 = \mathbf{H}\boldsymbol{h}_0 + \mathbf{R}^\top\boldsymbol{b},$$

and $(b)$ holds from the reverse triangle inequality.

When the irrelevant $\boldsymbol{h}_{\mathrm{ir}}$ has a small stable-ball radius,

$$\varepsilon(\boldsymbol{h}_{\mathrm{ir}}) < \frac{\|\boldsymbol{t} - \boldsymbol{h}_0\| - (\varepsilon(\boldsymbol{t}) + \varepsilon(\boldsymbol{h}_0))}{2\varepsilon(\boldsymbol{h}_0) + \varepsilon(\boldsymbol{t})}\varepsilon(\boldsymbol{h}_0),$$

and is close to $\boldsymbol{h}_0$ such that

$$\|\boldsymbol{h}_{\mathrm{ir}} - \boldsymbol{h}_0\| = \varepsilon(\boldsymbol{h}_{\mathrm{ir}}) + \varepsilon(\boldsymbol{h}_0),$$

we have

$$
\begin{aligned}
\|\mathbf{H}(\boldsymbol{h}_{\mathrm{ir}} - \boldsymbol{h}_0)\| &\leq \sigma_{\max}(\mathbf{H})\|\boldsymbol{h}_{\mathrm{ir}} - \boldsymbol{h}_0\| \\
&\leq \left(\frac{\varepsilon(\boldsymbol{t})}{\varepsilon(\boldsymbol{h}_0)} + 1\right)(\varepsilon(\boldsymbol{h}_{\mathrm{ir}}) + \varepsilon(\boldsymbol{h}_0)) \\
&\leq \left(\frac{\varepsilon(\boldsymbol{t}) + \varepsilon(\boldsymbol{h}_0)}{\varepsilon(\boldsymbol{h}_0)}\right) \\
&\quad \times \left(\varepsilon(\boldsymbol{h}_0)\frac{\|\boldsymbol{t} - \boldsymbol{h}_0\| - (\varepsilon(\boldsymbol{h}_0) + \varepsilon(\boldsymbol{t}))}{2\varepsilon(\boldsymbol{h}_0) + \varepsilon(\boldsymbol{t})} + \varepsilon(\boldsymbol{h}_0)\right) \\
&\overset{(a)}{<} \left(\frac{\varepsilon(\boldsymbol{t}) + \varepsilon(\boldsymbol{h}_0)}{\varepsilon(\boldsymbol{h}_0)}\right) \\
&\quad \times \left(\varepsilon(\boldsymbol{h}_0)\frac{\|\boldsymbol{t} - \boldsymbol{h}_0\| - (\varepsilon(\boldsymbol{h}_0) + \varepsilon(\boldsymbol{t}))}{\varepsilon(\boldsymbol{h}_0) + \varepsilon(\boldsymbol{t})} + \varepsilon(\boldsymbol{h}_0)\right) \\
&= \|\boldsymbol{t} - \boldsymbol{h}_0\| - (\varepsilon(\boldsymbol{h}_0) + \varepsilon(\boldsymbol{t})) \\
&\quad + (\varepsilon(\boldsymbol{h}_0) + \varepsilon(\boldsymbol{t})) \\
&= \|\boldsymbol{t} - \boldsymbol{h}_0\|
\end{aligned}
$$

where $(a)$ holds from the fact that $\varepsilon(\boldsymbol{h}_0) > 0$, so dropping one $\varepsilon(\boldsymbol{h}_0)$ in the denominator provides a valid upper bound. Therefore, we can safely remove the absolute value function in Eqn (†) and get

$$
\begin{aligned}
\|\Phi(\boldsymbol{h}_{\mathrm{ir}}) - \boldsymbol{h}_{\mathrm{ir}}\| &= \|\mathbf{H}\boldsymbol{h}_{\mathrm{ir}} + \mathbf{R}^\top \boldsymbol{b}\| \\
&\geq \big|\|\boldsymbol{t} - \boldsymbol{h}_0\| - \|\mathbf{H}(\boldsymbol{h}_{\mathrm{ir}} - \boldsymbol{h}_0)\|\big| \\
&= \|\boldsymbol{t} - \boldsymbol{h}_0\| - \|\mathbf{H}(\boldsymbol{h}_{\mathrm{ir}} - \boldsymbol{h}_0)\| \\
&\geq \|\boldsymbol{t} - \boldsymbol{h}_0\| - \sigma_{\max}(\mathbf{H})\|\boldsymbol{h}_{\mathrm{ir}} - \boldsymbol{h}_0\| \\
&\geq \|\boldsymbol{t} - \boldsymbol{h}_0\| - \left(\frac{\varepsilon(\boldsymbol{t})}{\varepsilon(\boldsymbol{h}_0)} + 1\right)(\varepsilon(\boldsymbol{h}_{\mathrm{ir}}) + \varepsilon(\boldsymbol{h}_0)) \\
&\geq \|\boldsymbol{t} - \boldsymbol{h}_0\| - \left(\frac{\varepsilon(\boldsymbol{t}) + \varepsilon(\boldsymbol{h}_0)}{\varepsilon(\boldsymbol{h}_0)}\right) \\
&\quad \times \left(\frac{\|\boldsymbol{t} - \boldsymbol{h}_0\| - (\varepsilon(\boldsymbol{h}_0) + \varepsilon(\boldsymbol{t}))}{2\varepsilon(\boldsymbol{h}_0) + \varepsilon(\boldsymbol{t})}\varepsilon(\boldsymbol{h}_0) + \varepsilon(\boldsymbol{h}_0)\right) \\
&= \|\boldsymbol{t} - \boldsymbol{h}_0\| - \left(\frac{\varepsilon(\boldsymbol{t}) + \varepsilon(\boldsymbol{h}_0)}{\varepsilon(\boldsymbol{h}_0)}\right) \\
&\quad \times \left(\frac{\|\boldsymbol{t} - \boldsymbol{h}_0\| - (\varepsilon(\boldsymbol{h}_0) + \varepsilon(\boldsymbol{t})) + 2\varepsilon(\boldsymbol{h}_0) + \varepsilon(\boldsymbol{t})}{2\varepsilon(\boldsymbol{h}_0) + \varepsilon(\boldsymbol{t})}\varepsilon(\boldsymbol{h}_0)\right) \\
&= \|\boldsymbol{t} - \boldsymbol{h}_0\| - (\varepsilon(\boldsymbol{t}) + \varepsilon(\boldsymbol{h}_0))\left(\frac{\|\boldsymbol{t} - \boldsymbol{h}_0\| + \varepsilon(\boldsymbol{h}_0)}{2\varepsilon(\boldsymbol{h}_0) + \varepsilon(\boldsymbol{t})}\right).
\end{aligned}
$$

Finally, it is easy to verify that this term is an upper bound of $\varepsilon(\boldsymbol{h}_{\mathrm{ir}})$, since

$$\|\Phi(\boldsymbol{h}_{\mathrm{ir}}) - \boldsymbol{h}_{\mathrm{ir}}\| - \varepsilon(\boldsymbol{h}_{\mathrm{ir}}) = \|\mathbf{H}\boldsymbol{h}_{\mathrm{ir}} + \mathbf{R}^\top \boldsymbol{b}\| - \varepsilon(\boldsymbol{h}_{\mathrm{ir}})$$

$$\overset{(a)}{\geq} \left( \|\boldsymbol{t} - \boldsymbol{h}_0\| - (\varepsilon(\boldsymbol{t}) + \varepsilon(\boldsymbol{h}_0)) \left( \frac{\|\boldsymbol{t} - \boldsymbol{h}_0\| + \varepsilon(\boldsymbol{h}_0)}{2\varepsilon(\boldsymbol{h}_0) + \varepsilon(\boldsymbol{t})} \right) \right)$$

$$- \left( \frac{\|\boldsymbol{t} - \boldsymbol{h}_0\| - (\varepsilon(\boldsymbol{h}_0) + \varepsilon(\boldsymbol{t}))}{2\varepsilon(\boldsymbol{h}_0) + \varepsilon(\boldsymbol{t})} \varepsilon(\boldsymbol{h}_0) \right)$$

$$= \frac{1}{2\varepsilon(\boldsymbol{h}_0) + \varepsilon(\boldsymbol{t})} \Big( \|\boldsymbol{t} - \boldsymbol{h}_0\| \, (2\varepsilon(\boldsymbol{h}_0) + \varepsilon(\boldsymbol{t}))$$

$$- (\|\boldsymbol{t} - \boldsymbol{h}_0\| + \varepsilon(\boldsymbol{h}_0))(\varepsilon(\boldsymbol{h}_0) + \varepsilon(\boldsymbol{t}))$$

$$- (\|\boldsymbol{t} - \boldsymbol{h}_0\| - (\varepsilon(\boldsymbol{h}_0) + \varepsilon(\boldsymbol{t}))) \, \varepsilon(\boldsymbol{h}_0) \Big)$$

$$= \frac{1}{2\varepsilon(\boldsymbol{h}_0) + \varepsilon(\boldsymbol{t})} \Big( \|\boldsymbol{t} - \boldsymbol{h}_0\| \, (2\varepsilon(\boldsymbol{h}_0) + \varepsilon(\boldsymbol{t}) - \varepsilon(\boldsymbol{h}_0) - \varepsilon(\boldsymbol{t}))$$

$$- (\varepsilon(\boldsymbol{h}_0)(\varepsilon(\boldsymbol{h}_0) + \varepsilon(\boldsymbol{t})) - \varepsilon(\boldsymbol{h}_0)(\varepsilon(\boldsymbol{h}_0) + \varepsilon(\boldsymbol{t}))) \Big)$$

$$= 0,$$

where $(a)$ applies the lower bound to the first term, and the upper bound to the second term. In conclusion, we have

$$\|\Phi(\boldsymbol{h}_{\mathrm{ir}}) - \boldsymbol{h}_{\mathrm{ir}}\| \geq \varepsilon(\boldsymbol{h}_{\mathrm{ir}}),$$

i.e., $\Phi(\boldsymbol{h}_{\mathrm{ir}}) \notin B(\boldsymbol{h}_{\mathrm{ir}}, \varepsilon(\boldsymbol{h}_{\mathrm{ir}}))$. This completes our proof. $\qquad\square$

## B.2 MORE DISCUSSIONS ON ASMP 2.1.

Our Thm 2.3 is built upon Asmp 2.1. Informally, It assumes that the knowledge can be generated if representation takes some specific value. While this assumption may not hold especially in challenging scenarios (see App A for more discussions), it is reasonable for Thm 2.3.

Particularly, The goal of Thm 2.3 is to reveal how *linearity* in ReFT can *inevitably* hurt locality, *even if it appears successful in editing*. Therefore, our focus is on cases *where ReFT is capable of conducting the edits*. The existence of such cases are confirmed by our experiments, and by its effectiveness in diverse post-training tasks as demonstrated in Wu et al. (2024). Presuming such a success, given that ReFT can only update representations, Asmp 2.1 assumes that by updating representations to some targeted (possibly unknown) value, ReFT steers output $y$ to convey the desired knowledge.

## B.3 PROOF OF LEM 2.4

**Lemma B.4.** *Let* $\mathbf{R} = [\boldsymbol{r}_1; \ldots; \boldsymbol{r}_r], \mathbf{A} = [\boldsymbol{a}_1, \ldots, \boldsymbol{a}_r], \boldsymbol{b}^\top = (b_1, \ldots, b_k)$, *and* $\mathbf{W}(\boldsymbol{h}) = diag(w_1(\boldsymbol{h}), \ldots, w_r(\boldsymbol{h}))$. *Then BaFT*

$$\Phi(\boldsymbol{h}) = \boldsymbol{h} + \sum_{k=1}^{r} w_k(\boldsymbol{h}) \boldsymbol{r}_k (\boldsymbol{a}_k^\top \boldsymbol{h} + b_k - \boldsymbol{r}_k^\top \boldsymbol{h}),$$

*can be expressed in a matrix form*

$$\Phi(\boldsymbol{h}) = \boldsymbol{h} + \mathbf{R}^\top \mathbf{W}(\boldsymbol{h}) \left( \mathbf{A}\boldsymbol{h} + \boldsymbol{b} - \mathbf{R} \right).$$

*When using constant weighting* $\mathbf{W}(\boldsymbol{h}) = \mathbf{I}$, *BaFT becomes to ReFT. Otherwise, rows of* $\mathbf{W}\mathbf{R}$ *are not orthonormal, making BaFT and ReFT nonequivalent.*

Table 5: Hyper-parameters of different methods. For baselines, we only provided settings that were different from Wang et al. (2024e).

| | HParams. | LLaMA 2-7b(-chat) Value | LLaMA 3-8b-Instruct Value | Gemma 1.1-7b-Instruct Value |
|---|---|---|---|---|
| FT-L | / | Following Wang et al. (2024e)'s recommendation for LLaMA 2. | | |
| ROME | / | Following Wang et al. (2024e)'s recommendation for LLaMA 2. | | |
| MEMIT | / | Following Wang et al. (2024e)'s recommendation for LLaMA 2. | | / |
| AdaLoRA | Maximum Steps | 70 for Single and Continual Editing; 200 for Batched Editing | | |
| GRACE | Maximum Steps | 100 | 250 | 100 |
| | Lay. to Interven | 27 | 27 | 24 |
| WISE$_{light}$ | Param. Updates | Restrict the original WISE logic to a randomly selected 1/8 area. | | |
| BaFT & ReFT | Subspace Rank | | 12 | |
| | Pos. to Intervene | | Last 3 of Input + Output | |
| | Lay. to Intervene | 9;18;24;28 | 9;18;24;28 | 18;20;22;24 |
| | Learning Rate | 3e-4 for Single and Continual Editing; 1e-4 for Batched Editing | | |
| | Maximum Steps | 40 for Single and Continual Editing; 70 for Batched Editing | | |
| | Locality Reg. (BaFT) | $\alpha = 0.01, \beta = 0.05, \gamma = 0.02$ | $\alpha = 0.01, \beta = 0.1, \gamma = 0.05$ | $\alpha = 0.01, \beta = 0.1, \gamma = 0.05$ |
| | Maximum Steps | 40 for Single and Continual Editing; 70 for Batched Editing | | |

*Proof.* The derivations essentially come from the fact that matrix product can be expressed by summation of outer products. In particular, we have

$$
\begin{aligned}
\Phi(\boldsymbol{h}) &= \boldsymbol{h} + \sum_{k=1}^{r} w_k(\boldsymbol{h}) \boldsymbol{r}_k (\boldsymbol{a}_k^\top \boldsymbol{h} + b_k - \boldsymbol{r}_k^\top \boldsymbol{h}) \\
&= \boldsymbol{h} + \left( \sum_{k=1}^{r} w_k(\boldsymbol{h}) \boldsymbol{r}_k \boldsymbol{a}_k^\top - \sum_{k=1}^{r} w_k(\boldsymbol{h}) \boldsymbol{r}_k \boldsymbol{r}_k^\top \right) \boldsymbol{h} + \sum_{k=1}^{r} w_k(\boldsymbol{h}) \boldsymbol{r}_k b_k \\
&= \boldsymbol{h} + \left( \mathbf{R}^\top \mathbf{W}(\boldsymbol{h}) \mathbf{A} - \mathbf{R}^\top \mathbf{W}(\boldsymbol{h}) \mathbf{R} \right) \boldsymbol{h} + \mathbf{R}^\top \mathbf{W}(\boldsymbol{h}) \boldsymbol{b} \\
&= \boldsymbol{h} + \mathbf{R}^\top \mathbf{W}(\boldsymbol{h}) \left( (\mathbf{A} - \mathbf{R}) \boldsymbol{h} + \boldsymbol{b} \right) \\
&= \boldsymbol{h} + \mathbf{R}^\top \mathbf{W}(\boldsymbol{h}) \left( \mathbf{A} \boldsymbol{h} + \boldsymbol{b} - \mathbf{R} \right),
\end{aligned}
$$

when $\mathbf{W}(\boldsymbol{h}) = \mathbf{I}$ takes the identity matrix, BaFT reduces to ReFT. This completes the proof. □

## C  IMPLEMENTATION DETAILS

We provide more implementation details about different methods.

Throughout all experiments, BaFT used a logistic regression for $w_k(\boldsymbol{h})$ for all $k \in [r]$. ReFT was implemented as a special case of BaFT with constant weight $\mathbf{W} = \mathbf{I}$. Load balancing loss and the optional Locality regularization were removed as they were defined for weights. In addition, BaFT and ReFT used the same optimizer AdamW (Loshchilov & Hutter, 2019) and learning rate. An early stopping was performed if the training loss is smaller than a pre-specified threshold 0.01. We also added this early stopping to AdaLoRA after observing a similar improvement. We kept encountering numeric issues when running MEMIT on Gemma, so we omitted these results.. Other hyper-parameters are reported in Tab 5.

## D  MORE EXPERIMENT RESULTS

### D.1  COMPLETE BATCHED CONTINUAL EDITING RESULTS

Here we report the complete Batched Editing results in Tab 6 and Tab 7 using batch size 10 and 50 respectively. The averaged result are shown in Fig 3. We noted that such a batched setting makes knowledge editing resemble more conventional continual learning (Miao et al., 2021; Chen et al., 2023; Wang et al., 2024c).

Table 6: Batched Editing performance on ZsRE dataset, evaluated after conducting $T$ times of editing with batch size 10 sequentially. Best Avg. results are in **bold** and second best are underlined.

| | $T = 1$ | | | | $T = 10$ | | | | $T = 100$ | | | |
|---|---|---|---|---|---|---|---|---|---|---|---|---|
| | Rel. | Gen. | Loc. | Avg. | Rel. | Gen. | Loc. | Avg. | Rel. | Gen. | Loc. | Avg. |
| LLaMA 2-7b | | | | | | | | | | | | |
| MEMIT | 0.89 | 0.84 | 0.97 | 0.90 | 0.87 | 0.82 | 0.93 | 0.87 | 0.04 | 0.04 | 0.02 | 0.03 |
| FT-L | 0.43 | 0.42 | 0.87 | 0.57 | 0.12 | 0.10 | 0.17 | 0.13 | 0.03 | 0.03 | 0.00 | 0.02 |
| AdaLoRA | 1.00 | 0.85 | 0.88 | **0.91** | 0.95 | 0.82 | 0.87 | 0.88 | 0.46 | 0.45 | 0.77 | 0.56 |
| ReFT | 0.94 | 0.86 | 0.86 | 0.89 | 0.92 | 0.83 | 0.86 | 0.87 | 0.64 | 0.60 | 0.76 | 0.67 |
| BaFT (Ours) | 0.93 | 0.84 | 0.95 | **0.91** | 0.92 | 0.83 | 0.95 | **0.90** | 0.59 | 0.55 | 0.98 | **0.71** |
| LLaMA 3-8b-Instruct | | | | | | | | | | | | |
| | Rel. | Gen. | Loc. | Avg. | Rel. | Gen. | Loc. | Avg. | Rel. | Gen. | Loc. | Avg. |
| MEMIT | 0.91 | 0.85 | 0.96 | **0.91** | 0.84 | 0.78 | 0.87 | 0.83 | 0.06 | 0.06 | 0.03 | 0.05 |
| FT-L | 0.33 | 0.32 | 0.53 | 0.39 | 0.01 | 0.01 | 0.00 | 0.01 | 0.01 | 0.01 | 0.00 | 0.01 |
| AdaLoRA | 1.00 | 0.88 | 0.76 | 0.88 | 0.94 | 0.83 | 0.75 | **0.84** | 0.34 | 0.34 | 0.75 | 0.48 |
| ReFT | 0.92 | 0.82 | 0.65 | 0.80 | 0.89 | 0.78 | 0.64 | 0.77 | 0.46 | 0.43 | 0.44 | 0.44 |
| BaFT (Ours) | 0.92 | 0.82 | 0.83 | 0.86 | 0.90 | 0.78 | 0.85 | **0.84** | 0.43 | 0.40 | 0.95 | **0.59** |
| Gemma 1.1-7b-Instruct | | | | | | | | | | | | |
| | Rel. | Gen. | Loc. | Avg. | Rel. | Gen. | Loc. | Avg. | Rel. | Gen. | Loc. | Avg. |
| FT-L | 0.04 | 0.04 | 0.02 | 0.03 | 0.01 | 0.01 | 0.00 | 0.01 | 0.01 | 0.01 | 0.00 | 0.01 |
| AdaLoRA | 1.00 | 0.87 | 0.83 | **0.90** | 0.93 | 0.81 | 0.82 | **0.85** | 0.34 | 0.34 | 0.59 | 0.42 |
| ReFT | 0.90 | 0.75 | 0.86 | 0.84 | 0.88 | 0.72 | 0.84 | 0.81 | 0.48 | 0.44 | 0.69 | 0.54 |
| BaFT (Ours) | 0.90 | 0.74 | 0.91 | 0.85 | 0.89 | 0.73 | 0.90 | 0.84 | 0.45 | 0.41 | 0.87 | **0.58** |

Table 7: Batched Editing performance on ZsRE dataset, evaluated after conducting $T$ times of editing with batch size 50 sequentially. Best Avg. results are in **bold**.

| | $T = 1$ | | | | $T = 10$ | | | | $T = 20$ | | | |
|---|---|---|---|---|---|---|---|---|---|---|---|---|
| | Rel. | Gen. | Loc. | Avg. | Rel. | Gen. | Loc. | Avg. | Rel. | Gen. | Loc. | Avg. |
| LLaMA 2-7b | | | | | | | | | | | | |
| MEMIT | 0.87 | 0.82 | 0.91 | **0.87** | 0.45 | 0.43 | 0.46 | 0.45 | 0.03 | 0.03 | 0.02 | 0.03 |
| FT-L | 0.39 | 0.39 | 0.63 | 0.47 | 0.13 | 0.10 | 0.07 | 0.10 | 0.07 | 0.05 | 0.02 | 0.05 |
| AdaLoRA | 1.00 | 0.86 | 0.76 | **0.87** | 0.78 | 0.69 | 0.79 | 0.75 | 0.51 | 0.51 | 0.76 | 0.59 |
| ReFT | 0.90 | 0.77 | 0.85 | 0.84 | 0.80 | 0.69 | 0.82 | 0.77 | 0.60 | 0.56 | 0.74 | 0.63 |
| BaFT (Ours) | 0.92 | 0.78 | 0.89 | 0.86 | 0.80 | 0.69 | 0.90 | **0.80** | 0.62 | 0.57 | 0.92 | **0.70** |
| LLaMA 3-8b-Instruct | | | | | | | | | | | | |
| | Rel. | Gen. | Loc. | Avg. | Rel. | Gen. | Loc. | Avg. | Rel. | Gen. | Loc. | Avg. |
| MEMIT | 0.88 | 0.83 | 0.89 | **0.87** | 0.45 | 0.42 | 0.46 | 0.44 | 0.00 | 0.00 | 0.04 | 0.01 |
| FT-L | 0.32 | 0.29 | 0.42 | 0.34 | 0.00 | 0.00 | 0.00 | 0.00 | 0.01 | 0.00 | 0.00 | 0.00 |
| AdaLoRA | 1.00 | 0.83 | 0.67 | 0.83 | 0.74 | 0.63 | 0.63 | 0.67 | 0.42 | 0.40 | 0.69 | 0.50 |
| ReFT | 0.92 | 0.74 | 0.52 | 0.73 | 0.75 | 0.61 | 0.49 | 0.62 | 0.48 | 0.42 | 0.43 | 0.44 |
| BaFT (Ours) | 0.92 | 0.75 | 0.62 | 0.76 | 0.75 | 0.61 | 0.72 | **0.69** | 0.49 | 0.43 | 0.78 | **0.57** |
| Gemma 1.1-7b-Instruct | | | | | | | | | | | | |
| | Rel. | Gen. | Loc. | Avg. | Rel. | Gen. | Loc. | Avg. | Rel. | Gen. | Loc. | Avg. |
| FT-L | 0.02 | 0.02 | 0.01 | 0.02 | 0.00 | 0.00 | 0.00 | 0.00 | 0.01 | 0.00 | 0.00 | 0.00 |
| AdaLoRA | 0.13 | 0.08 | 0.02 | 0.08 | 0.08 | 0.06 | 0.02 | 0.05 | 0.03 | 0.03 | 0.00 | 0.02 |
| ReFT | 0.88 | 0.68 | 0.79 | 0.78 | 0.71 | 0.57 | 0.72 | 0.67 | 0.48 | 0.42 | 0.61 | 0.50 |
| BaFT (Ours) | 0.87 | 0.68 | 0.87 | **0.81** | 0.72 | 0.57 | 0.83 | **0.71** | 0.50 | 0.45 | 0.81 | **0.59** |

## D.2 DOWNSTREAM LOCALITY PERFORMANCE

In this section we study how different editing methods affect the LLM's performance on unrelated downstream task, as an additional measure of locality. To this end, we follow Yao et al. (2023) and evaluate how the LLM's ability of answering PIQA questions from Bisk et al. (2020) that are unrelated to the editing. The correctness is measured by whether the LLM chooses the correct answer according to its perplexity. For more details we refer the readers to Yao et al. (2023).

Table 8: Downstream task (PIQA) performance after being edited with 100 ZsRE knowledge. LLM uses LLaMA-2.

|  | Base | AdaLoRA | FT-L | ROME | MEMIT | MELO | WISE$_{light}$ | ReFT | BaFT |
|---|---|---|---|---|---|---|---|---|---|
| PIQA Accu. | 0.77 | 0.48 | 0.75 | 0.5 | 0.52 | 0.77 | 0.77 | 0.77 | 0.77 |

## D.3 MORE DISCUSSION ON BaFT VS WISE

In our experiment, we note that BaFT achieves better parameter efficiency and speeds, at a cost of slightly lower performance, resulting in an *efficiency-effectiveness trade-off*. Notably, this efficiency of BaFT can be valuable in applications that require frequent knowledge updates.

In order to improve the effectiveness of BaFT, one possible solution is to use more parameters, given that BaFT parameter efficiency is already much higher than state-of-the-art baseline WISE. As discussed in Sec 3.3, when WISE's parameters number is reduced from WISE$_{full}$ WISE$_{light}$, its performance degrades drastically. In comparison, BaFT uses even much less parameters, but maintains a highly comparable performance. Given this, we expect that using better training hyper-parameters such as learning rate to make mild performance improvement, and more parameters are needed.

To validate this, we tried to add intervention to all layers (a common practice in ReFT (Wu et al., 2024)) and increase the subspace rank to 16. This made BaFT performance on editing LLaMA-2 with 100 ZsRE knowledge increased from 0.80 (Rel: 0.73, Gen: 0.68, Loc: 0.98) to 0.82 (Rel: 0.77, Gen: 0.73, Loc: 0.95). However, we noted that going higher subspace rank didn't help.

Therefore, we conjecture that to build larger BaFT (and ReFT), we need to incorporate sparsity on basis activation as well. This can help alleviate unintentional parameter updates as in GRACE (Hartvigsen et al., 2024) and WISE (Wang et al., 2024d). In addition, such a sparsity opens the door of automating position selections: as when all bases are inactivated, BaFT makes no updates on the representation, which is equivalent to dropping the position from the fine-tuning process. We plan to explore this direction in our future work.

