# OpenReview forum: "Unlocking Efficient, Scalable, and Continual Knowledge Editing with Basis-Level Representation Fine-Tuning"
_ICLR.cc/2025/Conference — ICLR 2025 Poster_

### Official Review · Reviewer_sAsi · 2024-10-28

**Soundness:** 2
**Presentation:** 3
**Contribution:** 2
**Rating:** 5
**Confidence:** 4

**Summary:**

This paper presents BaFT, a method designed to improve knowledge editing in large language models by fine-tuning representations instead of model parameters. Unlike traditional parameter-based approaches, BaFT performs selective edits at the basis level within a learned subspace, using an input-dependent weighting mechanism. This enables precise updates to targeted knowledge while preserving unrelated knowledge. Experimental results demonstrate that BaFT excels across multiple benchmarks in single, continual, and batched editing tasks, achieving efficient knowledge updates with fewer parameters. BaFT shows strong potential for effective, dynamic knowledge editing in real-world applications.

**Strengths:**

1.This paper introduces BaFT, which performs updates at the basis-vector level within a representation subspace, rather than relying on traditional parameter-based updates.

2.The paper provides substantial theoretical and empirical support for BaFT. The theoretical section thoroughly analyzes the limitations of existing linear update methods, offering a solid rationale for BaFT’s nonlinear representation fine-tuning approach.

3.BaFT offers a more efficient solution for knowledge editing in large language models, especially valuable in applications that require frequent knowledge updates.

**Weaknesses:**

1.Although the article presents the BaFT method and demonstrates its advantages in knowledge editing, the analysis of its limitations and potential flaws is relatively sparse.

2.The experiments in the article primarily focus on a few specific benchmark datasets, which may not comprehensively reflect the performance of BaFT in various real-world applications.

**Questions:**

1.The paper asserts that it is the first work to explore an alternative selective representation-based update for knowledge editing. However, to my knowledge, REMEDI[1], as an established editing method, has already conducted preliminary investigations in this area.

2.Assumption 2.2 has been proposed in GRACE[2]，authors should cite it.

3.Almost editing methods have examined the ‘locality’ of these methods to downstream tasks. I recommend that the authors incorporate an evaluation of downstream tasks in the paper to ensure the reliability of the proposed editing methods.

4.In the experimental section, I have concerns regarding the validity of the baseline evaluation methods. The authors did not specify the evaluation framework or the parameter choices employed. If the authors utilized the EasyEdit[3] framework to evaluate ROME and MEMIT, I suggest a unified re-evaluation of the baseline methods, as the EasyEdit evaluation framework has known mistakes.

5.In the context of continual(sequential) editing, BAFT exhibits slightly inferior performance compared to WISE, which is achieved at the cost of faster training and inference speeds. How should the trade-off between these two aspects be effectively balanced?

6.As an update representation for achieving editing, I believe this is not a reliable knowledge editing method compared to directly updating parameters.

7.There are several expression issues in the paper, such as "anLLM"; however, this did not influence my overall score.

[1].https://github.com/evandez/REMEDI

[2].https://arxiv.org/abs/2211.11031

[3].https://github.com/zjunlp/EasyEdit?tab=readme-ov-file

---

> ### Author Response · Authors · 2024-11-20
>
> We highly appreciate your effort and time spent reviewing our paper and thank you for your expertise and constructive comments. In the following, we address your comments and questions one by one, which will also be updated in the revised paper.
>
> >**Limitation and Potential issue of BaFT**
>
> We are pleased to see that our main contribution: (1) revealing how linear updates can lead to inherent editing-locality trade-off, and (2) how this inherent limitation motivates BaFT as an efficient non-linear generalization, are admired by the reviewer.
> Following the reviewer's suggestion,
> we updated our draft to include the following limitations and future improvement of BaFT:
>
> 1. The empirical success of BaFT was mainly established on standard benchmarks [1], which may not be sufficient to reflect the diverse real-world applications.
>
>
> 2. BaFT as a generalization of ReFT requires hyper-parameter tuning to determine proper positions and layers to add interventions. Our choice was selected based on recommended values from ReFT [2]. We plan to explore automating this process by imposing proper sparsity constraints on weights in our future work.
>
> 3. The promising performance of BaFT demonstrates its potential for efficient knowledge editing.
> However, it is still an open question if representation-based method is capable of fitting any editing (or updates) learnable for parameter-based methods. We plan to explore this direction in our future work.
>
> [1] EasyEdit. https://github.com/zjunlp/EasyEdit.
>
> [2] Wu et al. ReFT: Representation Finetuning for Language Models. 2024.
>
>
>
> >**The experiments which may not comprehensively reflect the performance of BaFT in various real-world applications.**
>
> Our experiment design follows previous works [1, 2].
> Following the reviewer's suggestion, we further include unrelated downstream task for additional locality evaluations (please see below for more details), and we showed that BaFT is again highly competitive with existing baselines while maintaining notable parameter efficiency.
> However, we agree that existing benchmarks are insufficient to reflect diverse real-world applications.
> In response, we discuses this as a limitation in the updated draft, and will explore knowledge editing using unstructured data or in the wild.
>
> [1] EasyEdit. https://github.com/zjunlp/EasyEdit.
>
> [2] Wang et al. WISE: Rethinking the Knowledge Memory for Lifelong Model Editing of Large Language Models. 2024.
>
>
> >**Previous work REMEDI**.
>
> We thank the reviewer for bringing this paper to our attention.
> After carefully reading through the paper, we agree that REMEDI is an important work that studied if knowledge can be updated in a pretrained model through the lens of modifying representations.
> We updated the "first work" claim in our draft accordingly.
> We noted that this does not affect the main contributions of our work (as summarized by the reviewer, being "the first" alone is not a key contribution). In addition, we noted the following difference between REMEDI and ours:
> 1. REMEDI relies on a large-scale dataset to train the model. This can be challenging to obtain especially in frequent knowledge editing, and is not assumed accessible in our setting.
> 2. REMEDI still applies a linear (affine) update and may suffer from the same linearity limitation.
>
>
> >**Assumption 2.2 has been proposed in GRACE.**
>
> We thank the reviewer for pointing out this connection.
> After rechecking the paper, we agree that the two assumptions share the same idea.
> We have updated the draft to highlight this.
>
> >**the "locality" of these methods to downstream tasks.**
>
> We agree with the reviewer that downstream tasks as a measure of to forgetting
> issue can provide useful information.
> Following the suggestion, we experiment the model performance on a commonsense dataset as in [1].
> Due to time constraint,
> we only tested continually editing LLaMA 2-7b with 100 ZsRE knowledge.
> The results are reported in the Table below and have been updated in the Appendix of our draft.
> We noted that our BaFT achieved competitive locality as state-of-the-art baselines.
>
> [1] Yao et al. Editing large language models: Problems, methods, and opportunities. 2023.
>
> | PIQA Accu. | Base | AdaLoRA | FT-L | ROME | MEMIT | MELO | WISE-light | ReFT | BaFT |
> |------------|------|---------|------|------|-------|------|-----------------------|------|------|
> |            | 0.77 | 0.48    | 0.75 | 0.5  | 0.52  | 0.77 | 0.77                  | 0.77 | 0.77 |

---

> ### Author Response · Authors · 2024-11-20
>
> >**Re-evaluate ROME and MEMIT**.
>
> We thank the reviewer for providing this valuable information.
> Yes our experiments were conducted based on EasyEdit.
> In response, we conducted a systematic re-evaluation of MEMIT and ROME following your suggestion.
> Due to time constraint, we only finished experiments on LLaMA-2 and Gemma. We have updated corresponding results in our draft (highlighted in red).
> We will update the draft as soon as other experiments are finished.
> According to our new results,
> **MEMIT and ROME still fall short in challenging sequential (and batched) scenarios, as observed in [1], and perform worse than our BaFT.**
>
> [1] Wang et al. WISE: Rethinking the Knowledge Memory for Lifelong Model Editing of Large Language Models. 2024.
>
> >**Balance efficiency and effectiveness trade-off.**
>
> The effectiveness of BaFT can be improved by using more parameters,
> given that their parameter efficiency is already much higher than WISE.
> When the latter's parameters number is reduced from "full" to "light", its performance degrades drastically.
> In comparison, BaFT uses even much less parameters, but maintains a highly comparable performance.
> Given this, we expect that using better training hyper-parameters such as learning rate to make mild performance improvement, and more parameters are needed.
> To validate this, we tried to add intervention to all layers (a common practice in ReFT [1]) and increase the subspace rank to 16. This made BaFT performance on editing LLaMA-2 with 100 ZsRE knowledge
> increased from 0.80 (Rel: 0.73, Gen: 0.68, Loc: 0.98) to 0.82 (Rel: 0.77, Gen: 0.73, Loc: 0.95).
> However, we noted that going higher subspace rank didn't help.
> Therefore, we conjecture that to build larger BaFT (and ReFT), we need to incorporate sparsity on basis activation as well.
> This can help alleviate unintentional parameter updates as in GRACE [1] and WISE [2].
> In addition, such a sparsity opens the door of automating position selections: as when all bases are inactivated, BaFT makes no updates on the representation, which is equivalent to dropping the position from the fine-tuning process.
> We plan to explore this direction in our future work.
>
>
> >**As an update representation for achieving editing, I believe this is not a reliable knowledge editing method compared to directly updating parameters.**
>
> Updating representations has proven effective in generic post-training as shown in [1],
> and in this work, we showed that it is also a promising direction for knowledge editing (after fixing the linearity limitation).
> However, we agree with these successes are not sufficient to draw a conclusive statement that representation-based methods are superior than parameter-based methods. Critically, it is still an open question if representation-based method is capable of fitting any editing learnable for parameter-based methods, and we plan to explore this direction in our future work.
> We included this discussion in our draft.
>
> [1] Wu et al. ReFT: Representation Finetuning for Language Models. 2024.
>
> >**Typos**
>
> Thank you for catching the error! We fixed this mistake and a few others in the revised draft. We didn't highlight these rewordings in the revised draft.

---

> > ### Comment · Reviewer_sAsi · 2024-11-21
> >
> > I'm delighted to see your response, and I'm happy to raise the score from three points to five points.

---

> > > ### Author Response · Authors · 2024-11-21
> > >
> > > Dear reviewer sAsi,
> > >
> > > We are pleased to see that our response are helpful for solving your concerns, and are grateful for your insightful suggestions to improve our paper and your rescoring.
> > > At the same time, please feel free to share with us any other suggestions if you find helpful for further improving the quality of our paper.
> > >
> > > Best regards,
> > >
> > > Authors

---

> > > > ### Author Response · Authors · 2024-11-27
> > > >
> > > > Dear Reviewer sAsi,
> > > >
> > > > Again, we sincerely appreciate your valuable feedback.
> > > > We are delighted to see that our previous response are helpful, and are grateful for your thoughtful rescoring.
> > > >
> > > > We wonder if there are any remaining concerns that leads to the "marginally below threshold" evaluation. If so,
> > > > we would be happy to see if there is any chance we can address them and further improve the quality of our work.
> > > >
> > > > Best regards,
> > > >
> > > > Authors

---

### Official Review · Reviewer_rQH9 · 2024-10-30

**Soundness:** 3
**Presentation:** 3
**Contribution:** 3
**Rating:** 8
**Confidence:** 4

**Summary:**

This paper addresses the challenge of updating specific knowledge within Large Language Models (LLMs) while preserving other unrelated knowledge. Traditionally, methods have attempted to update only a small number of parameters in specific layers of LLMs, but these methods often fail to maintain the precision required for effective knowledge editing due to the inherent trade-off between editing and locality. The authors propose a novel approach called Basis-level Fine-Tuning (BaFT), which builds upon the Representation Fine-Tuning (ReFT) method. Theoretical analysis supports the introduction of BaFT, highlighting the limitations of linear representation fine-tuning and suggesting that a more precise method could improve both the specificity and locality of knowledge edits.  Instead of applying a uniform linear update across the subspace, BaFT computes a unique weight for each basis vector depending on the input representation. This allows BaFT to handle different types of knowledge adaptively, potentially resolving the inherent tension between editing performance and locality. The experiments on several models under different settings demonstrate the effectiveness of the proposed model.

**Strengths:**

1. The prposed method obtains great performance under different knowledge editing settings including the single，continual and batch editing.
2. The motivation is clear and reasonable and the experiments consider different kind of knowledge dataset which makes the evaluation strong.

**Weaknesses:**

1. There are some details missing from the main part and I have listed in the question part.
2. The proof for the failure in the ReFT is reasonable, but why the loss used in RaFT can alleviate the problem is not clear. A proof here can make the method and theory complete and robust.

**Questions:**

1. Some details of the method are not clear and I'm wondering when conducting ReFT or BaFT, how to decide the position for the fine-tuning? Is it the same position for different knowledge? In addition, is the experiment conducted on all layers of the specific LLM? This may be the basic information in ReFT, but for readers, it's a bit confusing.
2. I agree that editing the knowledge in a single space would lead to the general-loc trade-off, but I think the assumption about the ball in is a bit subjective, it would be better to provide more related work here.
3. Some typos: L187, missing blank ‘anLM’.

---

> ### Author Response · Authors · 2024-11-20
>
> We highly appreciate your effort and time spent reviewing our paper and thank you for your expertise and constructive comments. In the following, we address your comments and questions one by one, which will also be updated in the revised paper.
>
> >**Implementation Details.**
>
> We implemented ReFT as a special case of BaFT, and set their shared parameters to the same values tuned based on a model basis using ReFT.
> One expectation is in batched editing --- as we used larger batch size here, we used smaller learning rates, and more iteration numbers.
> Therefore, when editing different knowledge, all the hyper-parameters are the same.
> We provide details from the question. For others, please refer to Table 5 in App B of our draft.
>
> - Position: Following ReFT [1], we tuned the position as a hyper-parameter.
> In execution, we chose to learn (and apply) to the last 3 tokens in the input, and all in the output. This is the same for different knowledge.
>
> - Layer: We altered layer 9; 18; 24; 28 of LLaMA-2 and LLaMA-3; and layer 18;20;22;24 of Gemma. These choices were made as a simple subset of recommended values from ReFT [1].
>
> Conceptually, BaFT should be less sensitive to the position choice, as it can assign unimportant bases with low weights, and setting all to 0 is equivalent to unchoosing this position.
> We plan to explore automate position and layer selection in BaFT in our future work.
>
> [1] Wu et al. ReFT: Representation Finetuning for Language Models. 2024.
>
>
> >**why the loss used in RaFT can alleviate the problem?**
>
> Conceptually, the proof of Thm 2.3, how ReFT suffers from the general-loc trade-off, heavily relies on the linearity of ReFT.
> In words, the linearity "guarantees" that locality will be affected.
> Our BaFT, on the contrary, is non-linear. This makes the above proof fail to apply, and this "guarantee" disappear.
>
> How BaFT alleviates the problem lies in not only its non-linearity, but also how it handles different data non-linearly,
> as non-linearity itself does not directly imply better trade-off.
> We take the insight that the failure of linear updates in ReFT stems from the fact that it applies to all inputs *indiscriminately*.
> Built upon this, **we learn to "downweight" the influence of (unneeded) updates on unrelated inputs, as formulated by the loss used in BaFT**.
> To be specific, BaFT conducts a fine-grained basis (directional)-level "downweighting". In Sec 3.4, we showed that doing a coarse subspace-level control can result in worse trade-off.
>
> Finally, we agree with the reviewer that a more in-depth theoretical analysis on the expressiveness of BaFT can be helpful, and and we will explore this direction in our future work.
>
> >**Assumption 2.2 is a bit subjective.**
>
> After checking the literature,
> we noted that a similar assumption has been made in GRACE [1].
> We have updated the draft accordingly.
>
> [1] Hartvigsen et al. Aging with GRACE: Lifelong Model Editing with Discrete Key-Value Adaptors. 2023.
>
> >**Typos**
>
> Thank you for catching the error! We fixed this mistake and a few others in the revised draft. We didn't highlight these rewordings in the revised draft.

---

> > ### Author Response · Authors · 2024-11-24
> > **We look forward to your response as the discussion period ends soon**
> >
> > Dear Reviewer rQH9,
> >
> > Thank you once again for your commitment to reviewing our paper and assisting us in improving our work. We would like to remind you that the discussion window will close soon, and we eagerly await your feedback.
> >
> > We have provided detailed explanations for each of your concerns. We would greatly appreciate it if you could review our responses and let us know if they fully or partially address your concerns. Any additional comments you may have would also be highly appreciated.
> >
> > Best regards,
> >
> > Authors

---

> > > ### Comment · Reviewer_rQH9 · 2024-11-25
> > > **Response to the rebuttal**
> > >
> > > Dear Authors,
> > >
> > > Thanks for your reply,
> > > I decided to raise my score.

---

> > > > ### Author Response · Authors · 2024-11-26
> > > >
> > > > Dear reviewer rQH9,
> > > >
> > > > We are glad to know that our responses addressed your concerns, and are grateful for your positive rating!
> > > > Please feel free to share with us any other suggestions if you find helpful for further improving the quality of our paper.
> > > >
> > > > Best regards,
> > > >
> > > > Authors

---

### Official Review · Reviewer_pjBe · 2024-11-03

**Soundness:** 3
**Presentation:** 3
**Contribution:** 2
**Rating:** 3
**Confidence:** 4

**Summary:**

Building on the ReFT, this paper introduces a novel method called Basis-level Representation Fine-tuning (BaFT) for editing knowledge within Large Language Models (LLMs) while preserving unrelated information. The authors perform a theoretical analysis that highlights the inherent limitations of existing approaches, particularly linear representation fine-tuning, which often necessitates a trade-off between editing effectiveness and the retention of unrelated knowledge (locality). To address these challenges, BaFT calculates the weight for each basis in the subspace based on the input representation, facilitating a more adaptive management of diverse knowledge types. The authors conducted experiments across three different LLMs and evaluated them against five editing benchmarks, demonstrating that BaFT surpasses existing methods in both editing performance and parameter efficiency. Ultimately, BaFT strikes a superior balance between integrating new knowledge and maintaining existing unrelated information.

**Strengths:**

1. BaFT introduces a non-linear, input-dependent weighting mechanism for basis-level representation fine-tuning, which is a significant departure from traditional parameter-based updates.

2. The authors demonstrate the effectiveness of BaFT through extensive experiments on three different LLMs and five editing benchmarks, showing superior performance in various scenarios.

3. This paper provides a detailed and clear explanation of the proposed method BaFT in conjunction with the theory.

**Weaknesses:**

1. Innovation and Improvement Effect: The improvements of the method presented in this paper compared to ReFT are limited. Although the authors propose the ReFT-based enhancement method BaFT, experimental results indicate that these improvements show weak effectiveness in knowledge editing benchmark, thus suggesting a lack of innovation in this work.

2. Questions of Method Applicability:
- In Assumption 2.1, the authors assume as follows: "Let text x encode s and r; text y generated by the LM will convey o if its intermediate representation takes some targeted value t." For instance, a sentences in the WikiData dataset: “The name of the country which Goursez Vreizh is associated with is []” is input into the model, and the generation probabilities are used to assess the accuracy of knowledge editing. Although multiple datasets are presented in this paper, these datasets exhibit high homogeneity in type.

- In contrast, there are more complex benchmark datasets in the field of knowledge editing, such as MQuAKE[1] and KEBench[2]. These datasets require inputting questions into the model and determining whether the generated responses contain the corresponding answers. For example, the question from MQuAKE, “Which sport is Dudley Town F.C. associated with?” is answered with, “Dudley Town F.C. is associated with the sport of association football.” The complexity of such questions is significantly higher than that of the datasets used in this paper, making them more aligned with real-world application scenarios and challenging the authors' assumptions in Assumption 2.1. The generality of the methods in this paper has been questioned.

[1] MQUAKE: Assessing Knowledge Editing in Language Models via Multi-Hop Questions

[2] Stable Knowledge Editing in Large Language Models

3. Choice of Baselines: The baselines in this paper are relatively limited. Key literature, including LTE [1], MELO [2], StaleKE [3], and InstructEdit [4], which are fine-tuning-based methods, are not discussed or compared, resulting in a lack of comprehensiveness regarding the current field of knowledge editing.

[1] Learning to edit: Aligning llms with knowledge editing.

[2] Melo: Enhancing model editing with neuron-indexed dynamic lora.

[3] Stable Knowledge Editing in Large Language Models

[4] InstructEdit: Instruction-based Knowledge Editing for Large Language Models

4. Performance of Baseline Results: The baseline results reported in the paper are significantly lower than those in existing literature, especially in the Batch Editing and Continual Editing settings, where the performance of the MEMIT method is notably inferior to previous reports. This starkly contrasts with results from several studies (e.g., [1, 2, 3]), casting doubt on the validity of this paper.

[1] Model Editing at Scale leads to Gradual and Catastrophic Forgetting

[2] MQUAKE: Assessing Knowledge Editing in Language Models via Multi-Hop Questions

[3] Learning to edit: Aligning llms with knowledge editing.

**Questions:**

1. Should anLM in line 187 be changed to an LM?

---

> ### Author Response · Authors · 2024-11-20
>
> We highly appreciate your effort and time spent reviewing our paper and thank you for your expertise and constructive comments. In the following, we address your comments and questions one by one, which will also be updated in the revised paper.
>
> >**The improvements of the method presented in this paper compared to ReFT are limited.**
>
> This work establishes a formal analysis on how fine-tuning representation can be applied to edit learned knowledge in LLMs.
> To this end, we first show that how *linearity* in existing representation fine-tuning can induce an *inherent* trade-off between *editing new knowledge* and *maintaining unrelated knowledge*. Built upon this, we proposed BaFT as a flexible framework to break linearity.
> Conceptually, BaFT allows us to control how different representations are edited in the subspace a case-by-case way.
> We would like to highlight BaFT does not directly target on a better *editing* performance (here refers to learning new knowledge):
> By design, BaFT and ReFT leverages a same-sized subspace to edit representations.
> ReFT only needs to fit new knowledge, while BaFT takes locality into consideration as well.
> Therefore, **the improvement of BaFT lies in how it achieves better locality without sacrificing editing capability**, as shown in our experiments.
>
> >**The Applicability of Asmp 2.1 is questioned on tasks involving multi-hop reasoning.**
>
> We thank the reviewer for the insightful comments on our Asmp 2.1.
> Our Thm 2.3, together with the two assumptions,
> is to reveal how *linearity* in ReFT can inevitably hurt locality, *even if it appears successful in editing*.
> Therefore, our focus is on cases *where ReFT is capable of conducting the edits*, given its effectiveness in diverse post-training tasks [1].
> Presuming such a success,
> given that ReFT can only update representations,
> we assume that *by updating representations to some targeted (possibly unknown) value, ReFT steers output $y$ to convey the desired knowledge*, as presented in Asmp 2.1.
>
> We have updated our draft to make this clear.
>
> [1] Wu et al. ReFT: Representation Finetuning for Language Models. 2024.
>
>
> >**Baseline LTE, MELO, Stable KE, and InstructEdit.**
>
> Following the reviwer's suggestion, we update the draft accordingly to incorporate and discuss these related works.
> We also add MELO (using EasyEdit [1], the benchmark used in this paper) for comparison. Due to time limitation, we only run experiments on LLaMA-2 and Gemma and update corresponding results in our draft (highlighted in red).
> As shown in the updated draft, we note that **the performance of MELO degraded faster than our BaFT in challenging sequential scenarios.**
>
> [1] EasyEdit. https://github.com/zjunlp/EasyEdit.
>
> >**MEMIT results in Batched and Sequential Scenarios.**
>
> We thank the reviewer for providing this valuable comment.
> Our experiments were conducted on EasyEdit [1],
> and we note that some updates on MEMIT and ROME implementation and evaluations had been made until very recently therein.
> In response, we conducted a re-evaluation of MEMIT and ROME.
> Due to time constraint, we only finished experiments on LLaMA-2 and Gemma. We have updated corresponding results in our draft (highlighted in red).
> Our results now match [2]. We will update the draft as soon as LLaMA-3 experiment is finished.
> According to our revised result,
> **MEMIT and ROME still fall short in challenging sequential (and batched) scenarios, as observed in [2], and perform worse than our BaFT.**
>
> [1] EasyEdit. https://github.com/zjunlp/EasyEdit.
>
> [2] Wang et al. WISE: Rethinking the Knowledge Memory for Lifelong Model Editing of Large Language Models. 2024.
>
> >**Typos**
>
> Thank you for catching the error! We fixed this mistake and a few others in the revised draft. We didn't highlight these rewordings in the revised draft.

---

> > ### Comment · Reviewer_pjBe · 2024-12-03
> >
> > Additionally, in the paper [1], it is mentioned that the SERAC and MEND methods, which demonstrate better Locality performance, were not reported in this paper.
> > Additionally, the Fluency metric provided in the paper [1] is also absent from this paper. Based on our experience, when using Eazyedit for evaluation, this metric is usually calculated alongside several other metrics reported in your paper. However, it seems that the authors intentionally overlooked this metric and the aforementioned methods, which is puzzling.

---

> ### Comment · Reviewer_pjBe · 2024-11-24
>
> Thank you for the detailed response to my questions. However, a major concern remains: the performance of BaFT appears to be significantly limited compared to other baselines, such as ReFT and WISE, even in terms of locality. Furthermore, the authors do not seem to provide a clear explanation for this in the experiments. For instance, it is unclear why BaFT's performance does not surpass that of the baselines.
>
> Additionally, some statements in the paper seem to contradict each other. For example:
>
> 1. "BaFT and ReFT used a subspace of the same rank, so its editing performance should be upper bounded by ReFT in this case," and
> 2. "BaFT was capable of maintaining a better editing-locality trade-off: it achieved better locality and portability than ReFT with no degradation of the editing effectiveness."
>
> In my opinion, these two statements cannot both be true.

---

> ### Author Response · Authors · 2024-11-24
>
> Dear reviewer pjBe,
>
> We are glad to see that our response helped address other concerns.
> In the following, we address these two comments and questions one by one, which will also be updated in the revised paper.
>
> >**The performance of BaFT appears to be significantly limited compared to other baselines, such as ReFT and WISE, even in terms of locality.**
>
> **Update**:
>
> *We realized that the reviewer might refer to Table 1 Singe Edit performance here.
> In this table, LLaMA-2 baseline results were taken from [1] and showed "higher locality" than our BaFT.
> However, we noted that these results are unreliable, and there has been a very recent correction update on [1].
> In response, we also re-evaluated all methods on LLaMA-3 and Gemma, and updated Table 1 in our draft accordingly.
> **According to the revised results, our BaFT in fact achieved the best (much better than the original) editing performance (AVG) and locality than baselines in all cases.***
>
> [1] Zhang et al. A Comprehensive Study of Knowledge Editing for Large Language Models. 2024
>
> --------
> According to Table 2 in our paper,
> **BaFT outperformed ReFT in all cases in achieving higher locality**.
> For the discussion convenience, we included these results from Table 2 below:
> BaFT achieved locality (Column Loc.) $\geq$ 0.92 in all cases, while ReFT's locality can fall below 0.7.
>
> The locality of BaFT is slightly lower than WISE, but as discussed in Sec 3.3 (Parameter Efficiency),
> WISE takes 10-20 times more parameters than BaFT, and had a much lower parameter efficiency
> (As an evidence, when WISE parameters number is reduced from "full" to "light", its performance degrades drastically).
> We added this discussion, together with potential directions to further improve BaFT by adding more parameters, in the revised draft.
>
>
> |--- | --- | |  |  | T = 1   | |  |  | T = 10   |  |  |  | T = 100 |  |  |  | T = 1000 |
> |------------|-------|--------|--------|--------|--------|---------|---------|---------|---------|-----------|-----------|-----------|-----------|------------|------------|------------|------------|
> | --- | --- | Rel. | Gen. | Loc. | **Avg.** | Rel. | Gen. | Loc. | **Avg.** | Rel. | Gen. | Loc. | **Avg.** | Rel. | Gen. | Loc. | **Avg.** |
> | LLaMA 2 | ReFT | 1.00 | 0.95 | 0.94 | 0.96 | 0.90 | 0.85 | 0.88 | 0.87 | 0.78 | 0.74 | 0.83 | 0.78 | 0.58 | 0.56 | 0.73 | 0.62 |
> |  | BaFT (Ours) | 1.00 | 0.94 | 0.97 | 0.97 | 0.89 | 0.84 | 0.97 | 0.90 | 0.75 | 0.70 | 0.98 | 0.81 | 0.63 | 0.60 | 0.98 | 0.74 |
> | LLaMA 3 | ReFT | 1.00 | 0.97 | 0.93 | 0.97 | 0.90 | 0.84 | 0.87 | 0.87 | 0.68 | 0.61 | 0.74 | 0.68 | 0.48 | 0.45 | 0.64 | 0.52 |
> |  | BaFT (Ours) | 1.00 | 0.95 | 0.96 | 0.97 | 0.89 | 0.82 | 0.95 | 0.89 | 0.70 | 0.64 | 0.93 | 0.76 | 0.50 | 0.49 | 0.93 | 0.64 |
> | Gemma  | ReFT | 1.00 | 0.86 | 0.91 | 0.92 | 0.92 | 0.81 | 0.81 | 0.85 | 0.66 | 0.58 | 0.69 | 0.64 | 0.50 | 0.46 | 0.65 | 0.54 |
> |  | BaFT (Ours) | 1.00 | 0.84 | 0.94 | 0.93 | 0.92 | 0.80 | 0.92 | 0.88 | 0.70 | 0.62 | 0.92 | 0.75 | 0.48 | 0.45 | 0.92 | 0.62 |
>
>
> >**Contradicted Statements.**
>
> We apologize for causing the confusion.
>
> In the two statements, both "**editing performance**" in the first statement and "**editing effectiveness**" and in the second should refer to "**editing success**" (reliability, Rel.), they do not take "locality" into account.
>
> Here we would like to provide a few more clarifications.
> The first statement, "**BaFT and ReFT used a subspace of the same rank, so its editing performance should be upper bounded by ReFT in this case...**", highlighted that
> BaFT and ReFT are assigned with a same-ranked subspace to edit representations.
> Conceptually, ReFT only needs to fit new knowledge, but BaFT takes locality into consideration as well.
> **This additional locality consideration is a regularizer** that restricts BaFT's behavior.
> In contrast, ReFT can learn the subspace freely.
> Therefore, at a colloquial level, we called unrestricted ReFT an "upper bound" of BaFT in terms of the editing success (reliability, Rel.).
>
> The second statement, "**..it achieved better locality and portability than ReFT with no degradation of the editing effectiveness**", conveys that, when we restrict BaFT's behavior on a fine-grained basis-level (through the locality regularizer), we can actually achieve better "locality" (and "portability") with minimally hurting its editing success (reliability, Rel.).
>
> We thank the reviewer again for pointing out these confusing statements.
> In response,
> we have revised the draft to make these statements clearer.

---

> > ### Author Response · Authors · 2024-11-26
> >
> > Dear Reviewer pjBe,
> >
> > Thank you once again for your commitment to reviewing our paper and assisting us in improving our work.
> >
> > As we approach the end of the paper revision period, we would greatly appreciate if you could review our responses and let us know if our clarification addresses your concerns about the model performance and our paper presentations.
> >
> > At the same time, any additional comments you may have would also be highly appreciated.
> >
> > Best,
> > Authors

---

> > > ### Comment · Reviewer_pjBe · 2024-11-27
> > >
> > > Thank you for your follow-up reply. However, I believe I will maintain my current score. While I noticed that you updated the results in Table 1, I am not in favor of making such updates to the performance of baselines or your own model during the rebuttal phase. This approach can undermine the credibility of the results in the eyes of the reviewers and create confusion about which results should be trusted. I understand this was likely not your intention, but I wanted to share my concerns.

---

> > > > ### Author Response · Authors · 2024-11-27
> > > >
> > > > Dear Reviewer pjBe,
> > > >
> > > > We want to emphasize that,
> > > > all experiment results updates made to the paper are to **correct known mistakes in the benchmark and baseline results**, and to **improve** the credibility and quality of our paper.
> > > >
> > > > Our experiments are based on EasyEdit, a widely-used benchmark in the community. In fact, many recent works, including StableKE, Learning to Edit, and InstructEdit that the reviewer mentioned, used EasyEdit.
> > > > We believe that running experiments on EasyEdit can provide reliable and believable results.
> > > >
> > > > However, as recognized by other reviewers, **Some mistakes in EasyEdit were located and fixed after ICLR submission deadline, and the correction were published during the rebuttal period.** (All these corrections and timelines, as detailed on EasyEdit Github, are public). Therefore, providing a timely updated version of our experiments, to **correct these known mistakes** is critical to **improve the credibility and quality of our paper**.
> > > >
> > > > To make the correction, in Table 1, we took LLaMA2 baseline results from the latest version (to correct the mistakes) provided by EasyEdit team's survey paper [1]. Other results were re-evaluated based on the latest version of EasyEdit, without any additional hyperparameter tuning or modifications. Based on the revised (**and correct**) results, our method outperformed baselines in all cases.
> > > >
> > > > We understand that updating results showing improvement because "we noted some bugs in our codes" sounds suspicious. However, **this is not our case**: we updated our results in response to the announcement from EasyEdit that **previous results should be updated** because they are wrong.
> > > >
> > > > Finally, we noted that there is no further comments on our clarification of BaFT vs ReFT. We are pleased to see that our updated statement are now clear.
> > > >
> > > > Best,
> > > >
> > > > Authors

---

> > > > > ### Comment · Reviewer_rQH9 · 2024-11-27
> > > > >
> > > > > Just my opinion. I think the author's update on the results is reasonable.
> > > > > I also noticed that the EasyEdit team has announced their bugs.
> > > > > The author's update would make the results more convincing.

---

> ### Comment · Reviewer_pjBe · 2024-12-03
>
> I also checked EasyEdit GitHub, while their announcement is
>
> > 2024-11-19, we update the Table 4 results in the paper "A Comprehensive Study of Knowledge Editing for Large Language Models" after optimizing certain methods (related to AdaLoRA) and fixing computational bugs (related to ROME and MEMIT) in the EasyEdit (More details in #427).  **These improvements have led to better results than before. ** We will continue updating this paper and welcome everyone to discuss and exchange ideas.
>
> I don't understand why these baseline results are lower than the first-release results in the paper in terms of locality.
>
> However, the primary reason for this score is not as previously stated. The main concern is that the performance of BaFT seems to be considerably constrained when compared to other benchmarks like ReFT, especially in terms of locality. While I acknowledge that the performance gap may have widened slightly after the results were updated, I'm puzzled as to why issues in EasyEdit only impact the baselines and not your method in terms of locality. Additionally, in the initial release and current version, the results did not effectively support the rationale for higher locality.

---

> ### Author Response · Authors · 2024-12-03
>
> >**(Main Concern) The performance of BaFT seems to be considerably constrained when compared to other benchmarks like ReFT, especially in terms of locality.**
>
> As reported in our previous response, BaFT is capable of achieving far more better locality than ReFT (Table 2).
> In Table 1, BaFT achieved slight but **consistently higher** locality than ReFT.
> Notably, in Table 1, **due to the lack of off-the-shelf locality data for regularization, BaFT does not incorporate Locality Regularization.**
> In Sec 3.4 we conducted a careful ablation study, showing that **both basis-level weighting and locality regularization are critical for improving locality**, and the basis-level weighting is capable of **alleviate the editing-locality trade-off** due to its fine-grained nature.
>
> >**Why these baseline results are lower than the first-release results in the paper in terms of locality. Why issues in EasyEdit only impact the baselines and not your method in terms of locality.**
>
> According to the revised results, the performance of **all methods, including baselines and ours, are "improved"**. On LLaMA-2, baseline methods showed "lower" locality compared to the previous version. However, their editing success and portability significantly increased, resulting in an increased average score. Similarly, on LLaMA-3 baseline methods had "higher" locality and average score.
>
> >**SERAC and MEND methods were not reported in this paper.**
>
> As mentioned in Sec 3.1, we choose baselines that "**do not require a larges-scale hard-to-access training data, or training additional models**". Since these assumptions of access to such a large-scale data and substantial computational resource are far more restricted than the assumptions made in other methods, making a direct comparison unfair.
> Therefore, we didn't include SERAC, which requires additional training an in-scope classifier and a separate LM serving as the editing model, and MEND, which requires additional training a HyperNet on large dataset.
>
>
> >**Fluency is not reported.**
>
> The primary focus of this work is on improving editing-locality trade-off especially in challenging scenarios such as where frequent updates are desired. As a result, **we follow recent works on this direction (to just name a few, GRACE, WISE, MELO) and focus on editing related metrics.**

---

### Official Review · Reviewer_v1Lr · 2024-11-04

**Soundness:** 3
**Presentation:** 3
**Contribution:** 2
**Rating:** 6
**Confidence:** 3

**Summary:**

The paper discusses the challenge of updating LLMs with new knowledge without disturbing the existing knowledge they hold. Traditional methods of parameter-based updates have limitations in achieving this, as they tend to affect all inputs globally. The study introduces a new method called BaFT which achieves a better balance between making necessary updates and preserving unrelated knowledge. The effectiveness of BaFT is demonstrated through experiments on several LLMs across different editing benchmarks, showing superior performance compared to previous methods.

**Strengths:**

1. This paper is well-written and easy to follow. It also provides a clear introduction of the Knowledge Editing task matter, enhancing its accessibility to a broad audience.

2. The paper addresses a significant problem within the field of Knowledge Editing, focusing on a pertinent limitation associated with current methodologies. The investigation into this particular trade-off is could potentially lead to substantial advancements in the field of Knowledge Editing.

3. The experimental design and execution are robust, utilizing well-chosen benchmarks to effectively demonstrate the proposed method’s validity.

**Weaknesses:**

1. This paper lacks visual illustrations such as diagrams or figures, which would significantly aid in the comprehension of the methods and results. I recommend the authors add figures to enhance the reader’s understanding and engagement with the content.

2. There is an absence of publicly shared code in the paper or on platforms such as OpenReview. This hinders the reproducibility of the proposed method. I would suggest sharing the code through an anonymous GitHub repository or similar platform. This would greatly aid other researchers and reviewers in replicating and understanding the research.


3. While the motivation behind the study is well-articulated, the methods deployed are somewhat conventional. This paper devotes considerable space to arguing for directly applying the ReFT method to the Knowledge Editing tasks faces limitations. However, the proposed solutions introduced by the authors do not add intriguing or novel aspects to the existing array of techniques in the field.

**Questions:**

Please refer to the previous section.

---

> ### Author Response · Authors · 2024-11-20
>
> We highly appreciate your effort and time spent reviewing our paper and thank you for your expertise and constructive comments. In the following, we address your comments and questions one by one, which will also be updated in the revised paper.
>
> >**This paper lacks visual illustrations such as diagrams or figures.**
>
> Following the reviewer's suggestion, we added an illustration figure of our BaFT in the updated draft in Appendix A. We will reorganize the material to put the figure in the main body in the future revised version.
>
> >**There is an absence of publicly shared code in the paper or on platforms such as OpenReview**.
>
> As advised by the reviewer, we share the implementations of our method on the Anonymous link [1]. Our codes are mainly based on EasyEdit [2] and Official implementation of ReFT [3].
>
> [1] https://anonymous.4open.science/r/baft-E782/
>
> [2] https://github.com/zjunlp/EasyEdit
>
> [3] https://github.com/stanfordnlp/pyreft
>
> >**While the motivation behind the study is well-articulated, the methods deployed are somewhat conventional.**
>
> We are glad that the inherent limitation of linear representation fine-tuning, which motivates our BaFT, is admired by the reviewer.
> Our formal analysis on how linearity in representation fine-tuning can limit its performance is a main contribution of this work.
> This analysis sheds light on other fundamental tasks that require *selective* updates, such as in machine unlearning.
> For the proposed BaFT, its merit lies in the flexibility to admit a variety of improving directions. As we did in Sec 3.4, having some bases share weights, BaFT can be seen as learning a mixture of (smaller) subspaces. If sparsity constraints are further imposed, BaFT can naturally benefit from recent advance in sparse mixture of experts, where each subspace becomes an "expert".
> BaFT may also help understand how LLMs see knowledge by scrutinizing what bases (subspaces) combinations are chosen for different knowledge. We plan to explore these directions in our future work.

---

> > ### Author Response · Authors · 2024-11-24
> > **We look forward to your response as the discussion period ends soon**
> >
> > Dear Reviewer v1Lr,
> >
> > Thank you once again for your commitment to reviewing our paper and assisting us in improving our work. We would like to remind you that the discussion window will close soon, and we eagerly await your feedback.
> >
> > We have provided detailed explanations for each of your concerns. We would greatly appreciate it if you could review our responses and let us know if they fully or partially address your concerns. Any additional comments you may have would also be highly appreciated.
> >
> > Best regards,
> >
> > Authors

---

> > ### Comment · Reviewer_v1Lr · 2024-11-25
> >
> > Thanks for the reply!
> >
> > I think my rating and score are both fair and supportive. Best of luck with your submission!

---

> ### Author Response · Authors · 2024-11-26
>
> Dear reviewer v1Lr,
>
> We are glad to know that our responses addressed your concerns, and are grateful for your positive rating! Please feel free to share with us any other suggestions if you find helpful for further improving the quality of our paper.
>
> Best regards,
>
> Authors

---

### Author Response · Authors · 2024-11-21
**Summary of revision on the paper, and we thank all reviewers for providing valuable feedbacks.**

Dear reviewers and Area Chairs:

We highly appreciate your time and effort have put into our paper. We have uploaded a revised draft with the following major updates. Revised texts have been colored in red.

1. Noting that some corrections had been made on the benchmark our experiments run on, we conducted a re-evaluation of two pivotal baselines ROME and MEMIT. We noted that new results are better than before, but still fall behind compared to our method and other continual editing methods (e.g., WISE).

2. We discussed a few more recent advance in knowledge editing, and added MELO as a new baseline. We noted that our BaFT is able to achieve better performance than MELO. We also included a new experiment on how different editing methods affect the LLM's performance on downstream tasks, as a new measure of "locality". Our method again achieved excellent performance compared with different baselines.

3. We added a workflow figure to demonstrate the idea behind BaFT. We also added a new limitation discussion. Due to page and time limitation these materials are placed in the appendix. But we will move them to the main body in the future revised version.

4. Some rewording and rephrasing of less accurate statements, typos were also fixed.

Thanks in advance for your time, and we are looking forward to hearing from you if you have any additional questions or concerns.

Best regards, Authors

---

### Author Response · Authors · 2024-11-24
**New revision on the paper, and we thank all reviewers for providing valuable feedbacks.**

Dear reviewers and Area Chairs:

Again, we appreciate your time and effort have put into our paper. We have uploaded a revised draft (v2) with the following new major updates. All revised texts are colored in red.

- We completed the re-evaluation of ROME and MEMIT, along with new experiments with MELO, on LLaMA3 (due to the aforementioned correction on the benchmark we used). We noted that new results are better than before, but still fall behind compared to our method. Our observations of how BaFT achieved excellent editing-locality performance at high efficiency remains valid.

- Some new rewording and rephrasing of less accurate statements, typos were also fixed.

Best regards,

Authors

---

### Author Response · Authors · 2024-11-27
**New revision (V3) on the paper, and we thank all reviewers for providing valuable feedbacks.**

Dear reviewers and Area Chairs:

We appreciate your time and effort have put into our paper. We have uploaded a revised draft (v3). In this version, we updated Table 1 Single Editing performance evaluation, given that [1], our benchmark, and the source where our baseline performance were taken from, had a recent update on these results (made on Nov 17th). **According to the revised results, our BaFT achieved the best (much better than the original) editing performance (i.e., reliability, portability, and locality) than all baselines in all cases.** All of our conclusions in the paper remain valid.

As before, we keep all revised texts during the rebuttal period in red.

Best regards,

Authors


[1] Zhang et al. A Comprehensive Study of Knowledge Editing for Large Language Models. 2024

---

### Meta-Review · Area_Chair_SL3v · 2024-12-20

**Metareview:**

This paper tackles the challenge of updating specific knowledge in Large Language Models (LLMs) while preserving unrelated knowledge. Traditional methods, which attempt to update a small number of parameters in specific layers, often struggle to maintain precision due to the trade-off between editing and locality. The authors introduce a novel method called Basis-level Fine-Tuning (BaFT), an extension of Representation Fine-Tuning (ReFT). Theoretical analysis shows the limitations of linear representation fine-tuning and suggests that BaFT, by computing a unique weight for each basis vector based on input representation, can improve both specificity and locality in knowledge edits.  Although the experimental improvement of the proposed method over ReFT is relatively modest, I think that introducing Representation Fine-Tuning into knowledge editing is a bold and innovative attempt. Furthermore, the authors have conducted an extensive set of experiments, which constitutes a valuable contribution. I recommend that the authors revise the paper to better emphasize the key technical contributions within the presented theorem.

**Additional Comments On Reviewer Discussion:**

During the discussion, there were some differences of opinion among the reviewers. However, the authors showed a very sincere attitude and supplemented their work with extensive experiments. Most reviewers held a positive opinion.

---

### Decision · Program_Chairs · 2025-01-22

Accept (Poster)